# Online Inventory Problems: Beyond the i.i.d. Setting with Online Convex Optimization

**Massil Hihat**[1,2], **Stéphane Gaïffas**[1,2], **Guillaume Garrigos**[1,2], **Simon Bussy**[1]

[1]LOPF, Califrais' Machine Learning Lab, Paris, France
[2]Université Paris Cité and Sorbonne Université, CNRS,
Laboratoire de Probabilités, Statistique et Modélisation, Paris, France
`{hihat, stephane.gaiffas, garrigos}@lpsm.paris, simon.bussy@califrais.fr`

## Abstract

We study multi-product inventory control problems where a manager makes sequential replenishment decisions based on partial historical information in order to minimize its cumulative losses. Our motivation is to consider general demands, losses and dynamics to go beyond standard models which usually rely on newsvendor-type losses, fixed dynamics, and unrealistic i.i.d. demand assumptions. We propose MaxCOSD, an online algorithm that has provable guarantees even for problems with non-i.i.d. demands and stateful dynamics, including for instance perishability. We consider what we call non-degeneracy assumptions on the demand process, and argue that they are necessary to allow learning.

## 1 Introduction

An inventory control problem is a problem faced by an inventory manager that must decide how much goods to order at each time period to meet demand for its products. The manager's decision is driven by the will to minimize a certain *regret*, which often penalizes missed sales and storage costs. It is a standard problem in operations research and operations management, and the reader unfamiliar with the topic can find a precise description of this problem in Section 2.

The classical literature of inventory management focuses on optimizing an inventory system with complete knowledge of its parameters: we know in advance the demands, or the distribution they will be drawn from. Many efforts have been put into characterizing the optimal ordering policies, and providing efficient algorithms to find them. See e.g. the Economic Order Quantity model [7], the Dynamic Lot-Size model [30] or the newsvendor model [1]. Nevertheless, in many applications, the parameters of the inventory system are unknown. In this case, the manager faces a joint learning and optimization problem that is typically framed as a *sequential decision making* problem: the manager bases its replenishment decisions on data that are collected over time, such as past demands (observable demand case) or past sales (censored demand case). The early attempts to solve these *online* inventory problems employed various techniques and provided only weak guarantees or no guarantees at all [23, 4, 6, 12].

With the recent advances in online learning frameworks such as online convex optimization (OCO), bandits, or learning with expert advice, the literature of learning algorithms for inventory problems took a great leap forward. There is currently a growing body of research aiming at solving various online inventory problems while providing strong theoretical guarantees in the form of *regret bounds* [9, 13, 24, 3, 34, 25, 35, 15, 32, 33, 31].

However, these works rely on mathematically convenient but unrealistic assumptions. The most common one being that the demands are assumed to be independent and identically distributed (*i.i.d.*) across time, which rules out correlations and nonstationarities that are common in real-world scenarios.

37th Conference on Neural Information Processing Systems (NeurIPS 2023).

Furthermore, these works focus on specific cost structures (typically the *newsvendor* cost) and inventory dynamics (like lost sales models with nonperishable products), which we detail in Section 2.2. The main goal of this paper is to go beyond these restrictions and consider general demand processes, losses and dynamics, in order to provide numerical methods backed with theoretical guarantees compatible with real-world problems.

To do so, we recast the online inventory problem into a new framework called Online Inventory Optimization (OIO), which extends OCO. Our main contribution is a new algorithm called MaxCOSD, which can be seen as a generalization of the Online Subgradient Descent method. It solves OIO problems with provable theoretical guarantees under minimial assumptions (see Section 4). Here is an informal version of our main statement:

**Theorem 1** (Informal version of Theorem 12)**.** *Consider an OIO problem that satisfies convexity and boundedness assumptions. Assume further that demands are not degenerate (see Assumption 10). Then, running MaxCOSD (see Algorithm 2) with adequate adaptive learning rates gives an optimal $O(\sqrt{T})$ regret, both in expectation and in high probability.*

Our main assumption is a *non-degeneracy* hypothesis on the demand process, which we present and discuss in Section 5. This assumption generalizes typical hypotheses made in the inventory literature, while not requiring the demand to be *i.i.d.*. We also show that this assumption is sharp, in the sense that the OIO cannot be solved without such assumption. Finally, in Section 6 we present numerical experiments on both synthetic and real-world data that validate empirically the versatility and performances of MaxCOSD. The appendices gather the proofs of all our statements.

This paper helps to bridge the gap between OCO and inventory optimization problems, and we hope that it will raise awareness of OCO researchers to this family of under-studied problems, while being of importance to the industry.

## 2   A general model for online inventory problems

In this section, we present a new but simple model allowing to study a large class of online inventory problems. Then, in a series of remarks we discuss particular instances of these problems and the limitations of our model.

### 2.1   Description of the model and main assumptions

In the following, $n \in \mathbb{N} = \{1, 2, \dots\}$ refers to the number of products and $\mathcal{Y} \subset \mathbb{R}_+^n$ denotes the *feasible set*. An *online inventory optimization* (OIO) problem is a sequential decision problem where an *inventory manager* interacts with an *environment* according to the following protocol.

First, the environment sets the initial inventory state to zero ($x_1 = \mathbf{0} \in \mathbb{R}^n$), and chooses (possibly random) *demands* $d_t \in \mathbb{R}_+^n$ and *losses* $\ell_t : \mathbb{R}^n \to \mathbb{R}$ for every time period $t \in \mathbb{N}$. Then, the interactions begin and unfold as follows, for every time period $t \in \mathbb{N}$:

  i) The manager observes the *inventory state* $x_t \in \mathbb{R}^n$, where $x_{t,i}$ encodes the quantity of the $i^{\text{th}}$ product available in the inventory.

  ii) The manager raises this level by choosing an *order-up-to level* $y_t \in \mathcal{Y}$ which satisfies the *feasibility* constraint:
$$y_t \succeq x_t. \tag{1}$$
  Then, the manager receives instantaneously $y_{t,i} - x_{t,i} \geq 0$ units of the $i^{\text{th}}$ product.

  iii) The manager suffers a loss $\ell_t(y_t)$ and observes a subgradient $g_t \in \partial \ell_t(y_t)$.

  iv) The environment updates the inventory state by choosing $x_{t+1} \in \mathbb{R}^n$ satisfying the following *inventory dynamical constraint*:
$$x_{t+1} \preceq [y_t - d_t]^+. \tag{2}$$

The goal of the manager is to design an online *algorithm* that produces feasible order-up-to levels $y_t$ which minimizes the cumulative loss suffered using past observations (past inventory states and subgradients). Let us emphasize here the fact that demands and losses are not directly observable. Throughout the paper, we make the following assumptions on the feasible set and the losses.

**Assumption 2** (Convex and bounded problem)**.**

   i) *(Convex and bounded constraint)* The feasible set $\mathcal{Y}$ is closed, convex, nonnegative ($\mathcal{Y} \subset \mathbb{R}_+^n$) and bounded: $\operatorname{diam} \mathcal{Y} \leq D$ for some $D \geq 0$.

   ii) *(Convex losses)* For every $t \in \mathbb{N}$, the loss function $\ell_t$ is convex.

   iii) *(Uniformly bounded subgradients)* There exists $G > 0$ such that, for all $t \in \mathbb{N}$, $y \in \mathcal{Y}$ and $g \in \partial \ell_t(y)$ we have $\|g\|_2 \leq G$.

Apart from the non-negativity assumption $\mathcal{Y} \subset \mathbb{R}_+^n$ which is specific to inventory problems, Assumption 2 is very common in online convex optimization [36].

**Definition 3** (Regret)**.** Given an horizon $T \in \mathbb{N}$, we measure the performance of an algorithm with the usual notion of *regret* which is defined as:

$$R_T = \sum_{t=1}^{T} \ell_t(y_t) - \inf_{y \in \mathcal{Y}} \sum_{t=1}^{T} \ell_t(y). \tag{3}$$

Observe that $R_T$ is possibly random, so we call its expectation the *expected regret* $\mathbb{E}\left[R_T\right]$.

Notice that in our model any constant strategy is feasible (see Lemma 26 in the appendix). Thus, the regret (3) is well-defined and can be interpreted as the difference between the cumulative loss incurred by the algorithm and that incurred by the best feasible constant strategy[1]. It is an easy exercise to see that under Assumption 2 we always have $R_T \leq DGT$ (see Lemma 27 in the appendix). Thus, our goal is to design algorithms with *sublinear* regret with respect to $T$, achieving $\mathbb{E}[R_T] \leq o(T)$. Note that some authors consider the equivalent notion of averaged regret $(1/T)R_T$. In this context, we would talk about *no-regret* algorithms.

## 2.2 Description of standard inventory models and limitations of our model

**Remark 4** (On the demands)**.** Demands are modeled by a stochastic process $(d_t)_{t \in \mathbb{N}} \subset \mathbb{R}_+^n$ with fixed in advance distribution. In other words, we consider a general oblivious model for demands where we make *no* assumptions of regularity, stationarity or independence. Thus, our model accommodates both *i.i.d.* [9, 3, 25, 32, 33, 31] and *deterministic* demands [13, 24, 34, 35, 15], while also allowing for correlations and nonstationarities that appear for instance in autoregressive models. However, we rule out strategic behaviors: this is not a game-theoretic model.

**Remark 5** (On the dynamic)**.** Inventory states $x_t$ are constrained by the inventory dynamical constraint (2) which links states, demands, and order-up-to levels. This constraint resembles the notion of *partial perishability* introduced in [9, Section 3.3] which imposes further that inventory states are non-negative. In the following, we present some standard dynamics that conform to our model. We warn the reader that we try here to simplify the vocabulary used in the scattered inventory literature, and that some authors [9, 3] may refer to these dynamics using different terms[2].

- **Stateless dynamic.** In stateless inventory problems, we assume that no product is carried over from one period to the other, *i.e.* $x_t = \mathbf{0}$. Observe that due to the non-negativity assumption $\mathcal{Y} \subset \mathbb{R}_+^n$, the feasibility constraint (1) is satisfied by any choice of $y_t \in \mathcal{Y}$ in stateless problems. This means that stateless inventory problems coincide with the usual online convex optimization (OCO) framework [36]. See Appendix D.1 for a discussion on the relationship between OCO and OIO. Any dynamic which is not stateless is called **stateful**, see below.

- **Backlogging dynamic.** In backlogging inventory problems, excess demand stays on the books until it is satisfied and inventory leftovers are carried over to the next period. It corresponds to set $x_{t+1} = y_t - d_t$. Notice that in this case the inventory state may be negative to represent backorders. This kind of dynamic has been widely studied in the context of classical inventory theory due to its linear nature, see e.g. [26, Chapter 4].

---

[1]In the context of inventory problems, these constant strategies are known under the name of (stationary) base-stock policies, $S-$policies, or order-up-to policies. See e.g. [26, Chapter 4].

[2]For instance the stateless dynamic is usually referred to as the "perishable" setting, and lost sales and backlogging dynamics may both be referred to as "nonperishable" settings.

- **Lost sales dynamic.** Assume that products are nonperishable, excess demand is lost and inventory leftovers are carried over to the next period. We refer to this case as the lost sales dynamic which corresponds to set $x_{t+1} = [y_t - d_t]^+$. See e.g. [9, 25].

- **Perishable dynamic.** In perishable inventory systems [19], newly ordered products are fresh units that have a fixed usable lifetime. To model such a dynamic, it is necessary to track the entire age distribution of the on hand inventory and to specify the stockout type (lost sales or backlogging) and the issuing policy (i.e. how items are issued to meet demand). For instance, [32] describes a perishable setting modeling a single product with first-in-first-out issuing policy, which satisfies our inventory dynamical constraint (2).

All those dynamics are what we call *deterministic* dynamics, in the sense that they take the form:

$$(\forall t \in \mathbb{N}) \quad x_{t+1} = X_t(y_1, d_1, \ldots, y_t, d_t), \tag{4}$$

where $X_t : (\mathcal{Y} \times \mathbb{R}_+^n)^t \to \mathbb{R}^n$ is a fixed in advance function satisfying $X_t(y'_1, d'_1, \ldots, y'_t, d'_t) \preceq [y'_t - d'_t]^+$ for all realizations $(\ell'_t)_{t \in \mathbb{N}}$, $(d'_t)_{t \in \mathbb{N}}$ and $(y'_t)_{t \in \mathbb{N}}$.

**Remark 6** (On the feasible set). The feasible set $\mathcal{Y}$ models fixed constraints on the order-up-to levels. Since it should satisfy Assumption 2.i), our framework does not allow for discrete sets as they appear in [15] for instance. This is one of the main limitations of our work. Typical choices include box constraints: $\mathcal{Y} = \prod_{i=1}^n [\underline{y}_i, \overline{y}_i]$, or capacity constraints [25]: $\mathcal{Y} = \{y \in \mathbb{R}_+^n \mid \sum_{i=1}^n y_i \leq M\}$. Note that in some instances, the parameters of the box constraint is dictated by some assumptions on the problem. For instance, it is assumed in [9, 32] that $\mathcal{Y} = [0, D]$ where $D$ is a known upper bound for an optimal constant policy.

**Remark 7** (On the losses). Losses $(\ell_t)_{t \in \mathbb{N}}$ are random functions drawn before the interactions start. As for demands, we consider an oblivious model for the losses which allows to handle both the *i.i.d.* case and the *deterministic* case. Most interestingly, losses can depend on the demands. This allows to consider the *newsvendor* loss, which writes $\ell_t(y) = c(y, d_t)$ with:

$$c(y, d) = \sum_{i=1}^n \left( h_i [y_i - d_i]^+ + p_i [d_i - y_i]^+ \right). \tag{5}$$

Here $h_i \in \mathbb{R}_+$ and $p_i \in \mathbb{R}_+$ are respectively the unit holding cost (a.k.a. overage cost) and unit lost sales penalty cost (a.k.a. underage cost) of product $i \in [n]$. The newsvendor loss satisfy assumptions 2.ii) and 2.iii) with $G = \sqrt{n} \max_{i \in [n]} \max\{h_i, p_i\}$. Our model also accommodates the newsvendor loss with time-varying unit cost parameters, as long as these remain bounded.

Because the losses are drawn before-hand, our model usually does not allow to incorporate costs that depend explicitly on the inventory states such as purchase costs, outdating costs, or fixed costs. However, there are exceptions: when the dynamic is lost sales, purchase costs can be included into our model, by considering the newsvendor loss onto which a cost transformation is applied (a.k.a. *explicit formulation* [26, Paragraph 4.3.2.4]). See also [9, 25] and the references therein.

**Remark 8** (On the observability structure and subgradients). To handle arbitrary losses, we required in our model that in addition to inventory states $x_t$, at least one subgradient $g_t \in \partial \ell_t(y_t)$ is revealed at each period. In the case of the newsvendor loss, this is less demanding than both the *observable demand* setting and the *censored demand* setting [3]. In the former, the demand $d_t$ is revealed instead of a subgradient $g_t$, meaning that the manager has complete information on the newsvendor loss $\ell_t = c(\cdot, d_t)$. In the latter, the sale $s_t := \min\{y_t, d_t\}$ is revealed, allowing the manager to compute a subgradient through the following formula (see Lemma 28 in Appendix B):

$$\left( h_i \mathbb{1}_{\{y_{t,i} > s_{t,i}\}} - p_i \mathbb{1}_{\{y_{t,i} = s_{t,i}\}} \right)_{i \in [n]} \in \partial \ell_t(y_t).$$

In this case, we see that no randomization is involved in the choice of the subgradient. This means that the subgradient selection is *deterministic*, in the sense that:

$$(\forall t \in \mathbb{N}) \quad g_t = \Gamma_t(\ell_t, y_t), \tag{6}$$

where $\Gamma_t : \mathbb{R}^{\mathbb{R}^n} \times \mathcal{Y} \to \mathbb{R}^n$ is a fixed in advance function such that $\Gamma_t(\ell'_t, y'_t) \in \partial \ell'_t(y'_t)$ for all realizations $(\ell'_t)_{t \in \mathbb{N}}$ and $(y'_t)_{t \in \mathbb{N}}$.

**Remark 9** (OIO vs OCO). In this final remark, we would like to point out that OIO is a novel and strict extension of OCO, that is, OIO cannot be casted into OCO or one of its known extensions. More details on this are provided in Appendix D.1.

# 3 Partial results for simple inventory problems

Previous works on online inventory problems mainly focused on two settings: stateless inventory problems and stateful inventory problems with *i.i.d.* demands. Both settings are discussed here.

## 3.1 Stateless inventory problems

In the literature of stateless inventory problems, arbitrary deterministic demands have already been considered. This has been done for instance in [13, 34, 35], which assume demand is observable and rely on the "learning with expert advice" framework. On the other hand [24] is the first work that considered the stateless setting with censored demand. See also [15] which tackled these problems in the discrete case, by reducing them to partial monitoring (an online learning framework that generalizes bandits). All these works achieve a $O(\sqrt{T})$ regret (up to logarithmic terms), but are restricted to the newsvendor cost structure.

In our work, we aim at solving inventory problems with arbitrary demands and losses. Recall that under Assumption 2, stateless inventory problems coincide with the standard OCO framework [36] since feasibility (1) is trivially verified (see Remark 5). Thus, a natural choice to solve stateless inventory problems is the Online Subgradient Descent (OSD) method, which we recall in Algorithm 1.

---

**Algorithm 1:** OSD

**1 Parameters:** learning rates $(\eta_t)_{t \in \mathbb{N}}$, initial order-up-to level $y_1 \in \mathcal{Y}$
**2 for** $t = 1, 2, \ldots$ **do**
**3**      Output $y_t$;
**4**      Observe $g_t \in \partial \ell_t(y_t)$;
**5**      Set $y_{t+1} = \mathrm{Proj}_{\mathcal{Y}}(y_t - \eta_t g_t)$;

---

Classical regret analysis of OSD (this is essentially proven in [8, Theorem 3.1], see also Corollary 20 in the appendix) shows that taking *decreasing* learning rates of the form $\eta_t = \gamma D/(G\sqrt{t})$ where $\gamma > 0$, leads to a regret bound $R_T = O(GD\sqrt{T})$. It must be noted that the $O(\sqrt{T})$ scaling is *optimal* under Assumption 2 (see e.g. [22, Theorem 5] or [21, Theorem 5.1]).

## 3.2 Stateful i.i.d. inventory problems

When non-trivial dynamics are involved, inventory problems are much more complex and have mainly been studied in the *i.i.d.* demands framework. We review here the literature of joint learning and inventory control with censored demand and refer to the recent review of [5, Chapter 11] for further references. We stress that all those papers obtain rates for the *pseudo-regret*, a lower bound of the expected regret which we consider in this paper (see Appendix D.2 for more details).

The seminal work of Huh and Rusmevichientong [9] is the first that derives regret bounds for the single-product *i.i.d.* newsvendor case under censored demand, it is also the sole work that considers general dynamics through their notion of *partial perishability* [9, Section 3.3]. They were able to design an algorithm called Adaptive Inventory Management (AIM) based on a subgradient descent and dynamic projections onto the feasibility constraint (1) which achieves a $O(\sqrt{T})$ pseudo-regret. Their main assumption is that the demands should not be degenerate in the sense that $\mathbb{E}[d_1] > 0$ and that the manager should know a lower bound $0 < \rho < \mathbb{E}[d_1]$. This lower bound is then used in AIM to tune adequately the learning rate of the subgradient descent. Their analysis is based on results from queuing theory which rely heavily on the *i.i.d.* assumption.

Shi et al. [25] designed the Data-Driven Multi-product (DDM) algorithm which extend the AIM method of [9] to the multi-product case under capacity constraints. They also derived a $O(\sqrt{T})$ pseudo-regret bound by assuming further that demands are pairwise independent across products and that $\mathbb{E}[d_{1,i}] > 0$ for all $i \in [n]$ amongst other regularity conditions.

Zhang et al. [32] tackled the case of single-product *i.i.d.* perishable inventory systems with outdating costs. They designed the Cycle-Update Policy (CUP) algorithm which updates the order-up-to level according to a subgradient descent, but only when the system experiences a stockout, *i.e.* when $x_t = 0$, the order-up-to level remains unchanged otherwise. Feasibility is guaranteed using such

a policy. However, in order to derive $O(\sqrt{T})$ pseudo-regret they need to ensure that the system experiences frequently stockouts. To do so, they consider a stronger form of non-degeneracy, namely, $\mathbb{P}[d_1 \geq D] > 0$ where $\mathcal{Y} = [0, D]$.

Variants of the subgradient descent have also been developed in order to achieve $O(\sqrt{T})$ pseudo-regret in inventory systems that includes lead times [33] or fixed costs [31] which are both beyond the scope of our model.

To summarize, an optimal $O(\sqrt{T})$ rate for the pseudo-regret is achievable in many stateful inventory problems, under the *i.i.d.* assumption. To prove so, most of the cited works developed specific variants of the subgradient descent that accommodates the specific dynamic at play. We will show in Section 4 that this optimal $O(\sqrt{T})$ rate can be achieved by our algorithm MaxCOSD when applied to general inventory problems, with no *i.i.d.* assumption on the demand.

# 4 MaxCOSD: an algorithm for general inventory problems

In this section, we introduce and study our main algorithm: the Maximum Cyclic Online Subgradient Descent (MaxCOSD) algorithm. It is a variant of the subgradient descent where instead of changing the order-up-to levels at every period, updates are done only at certain *update periods* denoted $(t_k)_{k \in \mathbb{N}}$. These define *update cycles* $\mathcal{T}_k = \{t_k, \ldots, t_{k+1} - 1\}$ during which the order-up-to level remains unchanged: $y_t = y_{t_k}$ for all $t \in \mathcal{T}_k$. Update periods are dynamically triggered by verifying, at the beginning of each time period $t \in \mathbb{N}$, whether a *candidate order-up-to level* $\hat{y}_t$ is feasible or not. This candidate is computed by making a subgradient step in the direction of the subgradients accumulated during the cycle and using the following *adaptive* learning rates,[3]

$$\eta_t = \frac{\gamma D}{\sqrt{\left\|\sum_{s=t_k}^t g_s\right\|_2^2 + \sum_{m=1}^{k-1}\left\|\sum_{s \in \mathcal{T}_m} g_s\right\|_2^2}} \quad \text{for all } t \in \mathcal{T}_k. \tag{7}$$

The pseudo-code for MaxCOSD is given in Algorithm 2.

---
**Algorithm 2:** MaxCOSD

**1 Parameters:** learning rate parameter $\gamma > 0$, initial order-up-to level $y_1 \in \mathcal{Y}$
**2 Initialization:**
**3**     Set $\hat{y}_1 = y_1$, $t_1 = 1$, $k = 1$;
**4 for** $t = 1, 2, \ldots$ **do**
**5**     Output $y_t$;
**6**     Observe $g_t \in \partial \ell_t(y_t)$ and $x_{t+1}$;
**7**     Compute $\hat{y}_{t+1} = \text{Proj}_{\mathcal{Y}}\left(\hat{y}_{t_k} - \eta_t \sum_{s=t_k}^t g_s\right)$ where $\eta_t$ is defined[3] in Eq. (7);
**8**     **if** $x_{t+1} \preceq \hat{y}_{t+1}$ **then**
**9**         Set $y_{t+1} = \hat{y}_{t+1}$, $t_{k+1} = t+1$, $k = k+1$;
**10**     **else**
**11**         Set $y_{t+1} = y_t$;

---

MaxCOSD is inspired by CUP [32], which handles the feasibility constraint through cyclical updates. This approach differs for instance from AIM [9] or DDM [25], where the feasibility is enforced through projections onto the constraint. Among the differences between MaxCOSD and CUP is the fact that their cycle definition differ: in CUP stockouts trigger updates, whereas MaxCOSD relies directly on the feasibility condition, making its updates more frequent. Also, we use adaptive learning rates inspired by AdaGrad-Norm learning rates [27, Theorem 2] which allows us to be adaptive to the constant $G$ and obtain high probability regret bounds which are not available for CUP. Finally, and most importantly, the assumptions required by CUP are restrictive: *i.i.d.* demands, single-product, perishable dynamic and a strong form of a demand non-degeneracy. On the other hand, MaxCOSD performs well under much milder assumptions, which we introduce next.

---
[3]It may happen that $\eta_t$ is undefined due to a denominator that is zero in Eq. (7), in such cases set $\eta_t = 0$.

**Assumption 10** (Uniformly probably positive demand). There exists $\mu \in (0, 1]$ and $\rho > 0$ such that, for all $t \in \mathbb{N}$, almost surely,

$$\mathbb{P}\left[\forall i \in [n],\ d_{t,i} \geq \rho \mid \ell_1, d_1, \ldots, \ell_{t-1}, d_{t-1}\right] \geq \mu. \tag{8}$$

In simple settings we recover through Assumption 10 conditions that already appeared in the literature:

- In single-product *i.i.d.* newsvendor inventory problems, our assumption is equivalent to the existence of $\rho > 0$ such that $\mathbb{P}[d_1 \geq \rho] > 0$, that is, $\mathbb{P}[d_1 > 0] > 0$, or equivalently $\mathbb{E}[d_1] > 0$. This is exactly the non-degeneracy assumption required by [9] in AIM.
- In its multi-product extension, [25] assumes also pairwise independence across products. If we rather require mutual independence across products, then, Assumption 10 rewrites $\mathbb{P}[d_{1,1} > 0] \cdots \mathbb{P}[d_{1,n} > 0] > 0$, thus, our assumption reduces to $\mathbb{E}[d_{1,i}] > 0$ for all $i \in [n]$ which is also required by [25] in their algorithm DDM.
- We recover an assumption made for CUP [32] by requiring $\rho = D$ where $\mathcal{Y} = [0, D]$, and where $D$ is also assumed to be large enough (see Remark 6).
- If the demand is deterministic, then $\mu = 1$ and Eq. (8) becomes $d_{t,i} \geq \rho$ for all $i \in [n]$.
- If the demand is discrete, i.e. $d_{t,i} \in \{0, 1, \ldots\}$ for all $t \in \mathbb{N}, i \in [n]$, then, we can take $\rho = 1$ and rewrite Eq. (8) as follows: $\mathbb{P}[\exists i \in [n], d_{t,i} = 0 | \ell_1, d_1, \ldots, \ell_{t-1}, d_{t-1}] \leq 1 - \mu$.

In addition to Assumption 10 we also introduce a mild technical condition.

**Assumption 11** (Deterministic dynamic and subgradient selection). Dynamics and subgradient selections are deterministic, see Eq. (4) and Eq. (6).

We can now state our main result: under this new set of assumptions MaxCOSD achieves an optimal $O(\sqrt{T})$ regret bound both in expectation and in high probability.

**Theorem 12.** *Consider an inventory problem, and let assumptions 2, 10 and 11 hold. Then, Max-COSD (see Algorithm 2) run with $y_1 \in \mathcal{Y}$ and $\gamma > 0$ is feasible. Furthermore, when $\gamma \in (0, \rho/D]$, it enjoys the following regret bounds for all $T \in \mathbb{N}$,*

$$\mathbb{E}[R_T] \leq \frac{\sqrt{2}GD}{\mu}\left(\frac{1}{2\gamma} + \gamma + 1\right)\sqrt{T},$$

*and for any confidence level $\delta \in (0, 1)$ we have with probability at least $1 - \delta$,*

$$R_T \leq GD\left(\frac{1}{2\gamma} + \gamma + 1\right)\left(1 + \frac{1}{\mu}\log\left(\frac{T}{\delta}\right)\right)\sqrt{T}.$$

The obtained regret scales in $O(\sqrt{T})$, which is similar to other methods in the literature (see Section 3.2). As for the scaling in the constants $\mu$ and $\rho$, our rate scales like $O(\frac{1}{\rho\mu})$ and we do not know whether this can be improved or not. As a matter of comparison, CUP enjoys the same scaling (see [32, Theorem 2 & Remark 3 & Assumption 1]).

## 5   Non-degenerate demands are needed for stateful inventory problems

Throughout this paper, we have seen instances of OIO which can be solved with sublinear regret rates: stateless OIO (equivalent to OCO), and some stateful OIO. It must be noted that, contrary to OCO, solving those stateful OIO problems required a non-degeneracy assumption on the demand (see Assumption 10 and Section 3.2). We argue here that such an assumption is necessary for solving stateful OIO. Note that this idea is not new, and was already observed in the conclusion of [3]: *"To control for the impact of overordering, demands must be bounded away from zero, at least in expectation."*. Our contribution is to make this observation formal.

**Proposition 13.** *Given any feasible deterministic[4] algorithm for the single-product lost sales newsvendor inventory problem with observable demand over $\mathcal{Y} = [0, D]$, there exists a sequence of demands such that the regret is linear, i.e. $R_T = \Theta(T)$.*

---

[4] In general, a deterministic algorithm is defined by fixed in advance functions $Y_t : (\mathbb{R}^n \times \mathbb{R}^n)^{t-1} \to \mathcal{Y}$ and outputs $y_t = Y_t(g_1, x_2, \ldots, g_{t-1}, x_t)$ for all $t \in \mathbb{N}$.

Proposition 13 shows that Assumption 2 is not sufficient to reach sublinear regret in general inventory problems. This is totally unusual from an OCO perspective, and is a specificity of *stateful* OIO. Furthermore, the above result shows that what prevents us from reaching sublinear rates is not the limited feedback (demands and losses are observable in this example) but rather *zero demands*. This is why it is necessary to make an assumption preventing demands to be too small, in some sense. Note that one may think of imposing positive demands to circumvent this difficulty, but this is not sufficient. Indeed, a sequence of demands converging too fast to zero can also be problematic.

**Proposition 14.** *Given any feasible algorithm for the single-product lost sales problem with observable demand over $\mathcal{Y} = [0, D]$ such that $y_1 \in (0, D)$, there exists a constant sequence of losses and a sequence of* positive *demands such that the regret is linear, i.e. $R_T = \Theta(T)$.*

Let us now investigate why degenerated demand becomes a problem when going from stateless to stateful OIO. The main difference between the two is that the feasibility constraint is always trivially satisfied for stateless OIO (see Remark 5). Instead, for stateful OIO, we can show that the higher is the demand, the easier it is for the feasibility constraint to be satisfied.

**Lemma 15.** *Let $y, y', d \in \mathbb{R}_+^n$. If $\|y' - y\|_2 \leq \min_{i \in [n]} d_i$, then, $y' \succeq [y - d]^+$. In particular, given an inventory problem and a time period $t \in \mathbb{N}$, taking $y_{t+1} \in \mathcal{Y} \subset \mathbb{R}_+^n$ such that $\|y_{t+1} - y_t\|_2 \leq \min_{i \in [n]} d_{i,t}$ ensures that $y_{t+1}$ is feasible, in the sense that $x_{t+1} \preceq y_{t+1}$.*

The above lemma shows that if $y_{t+1}$ is taken close enough from the previous $y_t$, then the algorithm is feasible. The key point here is that "close enough" is controlled by the demand, meaning that when the demand is closer to zero there are less feasible choices for the manager. In such a case, we understand that it may be impossible to achieve sublinear regret, because the set of feasible choices could be too reduced.

The distance between two consecutive decisions can easily be controlled in methods based on subgradient descents, through their learning rates. This is why Lemma 15 is very helpful in the design of efficient feasible algorithms. It has been employed in the proof of our main result regarding MaxCOSD (Theorem 12). In the following, we further illustrate its usefulness by showing that OSD (see Algorithm 1) with adequate learning rates is feasible when the demand is *uniformly positive*.

**Assumption 16** (Uniformly positive demand). There exists $\rho > 0$ such that for all $t \in \mathbb{N}$, $i \in [n]$,

$$d_{t,i} \geq \rho. \tag{9}$$

**Theorem 17.** *Consider an inventory problem, and let assumptions 2 and 16 hold. Then, OSD (see Algorithm 1) run with $y_1 \in \mathcal{Y}$ and $\eta_t = \gamma D/(G\sqrt{t})$ where $\gamma \in (0, \rho/D]$, is feasible and satisfies for all $T \in \mathbb{N}$ that $R_T \leq (1 + 2\gamma)(2\gamma)^{-1} GD\sqrt{T}$.*

## 6 Numerical results

The goal of the following numerical experiments[5] is to show the versatility and performances of MaxCOSD in various settings. Let us consider the following problems.

- **Setting 1.** Single-product lost sales inventory problem with *i.i.d.* demands drawn according to Poisson(1).

- **Setting 2.** Single-product perishable inventory problem with a lifetime of 2 periods and *i.i.d.* demands drawn according to Poisson(1).

- **Setting 3.** Multi-product lost sales inventory problem with $n = 100$ and capacity constraints. Demands are *i.i.d.* and drawn independently across products according to Poisson($\lambda_i$) where the intensities $\lambda_i$ have been drawn independently according to Uniform[1, 2].

- **Setting 4.** Multi-product lost sales inventory problem with $n = 3049$ and capacity constraints. Demands are taken from the real-world dataset of the M5 competition [17].

- **Setting 5.** Multi-product lost sales inventory problem with $n = 3049$ and box constraints. As in Setting 4 we considered demands from the M5 competition dataset [17].

---

[5]The code is available at https://github.com/Califrais/newsvendor_tester.

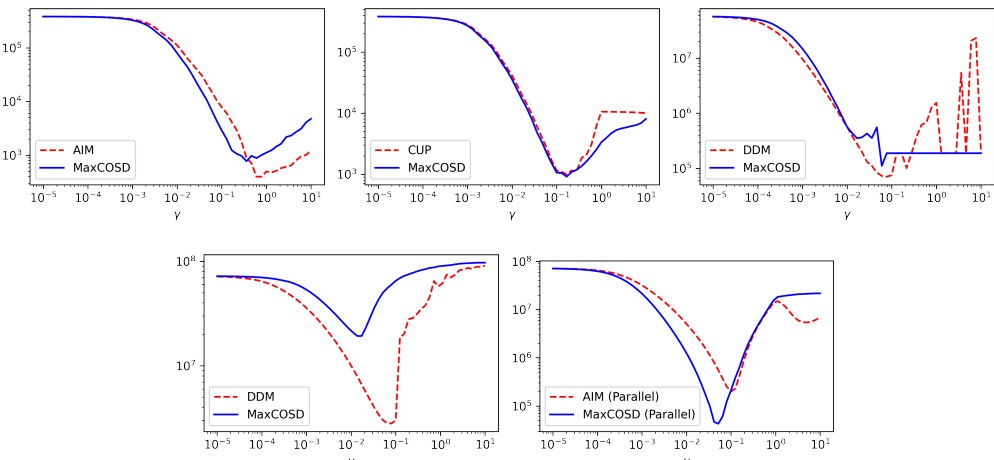

Figure 1: Regret in settings 1 to 5 (from left to right) as a function of the learning rate parameter $\gamma$.

We use the newsvendor loss in all the settings. In all the settings the cost parameters satisfy $p_i/h_i = 200$ since this ratio is known to exceed 200 in many applications [10]. In settings 1, 2 and 3, $h_i = 1$ and in settings 4 and 5, $h_i$ and $p_i$ are proportional to the real average selling costs.

In settings 1, 2, 3 and 4 we compare MaxCOSD against the following baselines: AIM [9] for Setting 1, CUP [32] for Setting 2, and DDM [25] for settings 3 and 4. In Setting 5, we ran a parallelized version of MaxCOSD, that is, one instance of MaxCOSD per product, and a parallelized version of AIM. Notice that in Settings 4 and 5 demands are not *i.i.d.* and thus, do not fit the assumptions of the baselines considered. Notice also that Theorem 12 requires MaxCOSD's learning rate to be small enough ($\gamma \leq \rho/D$), which we do not try to enforce here. All the algorithms have been initialized with $y_1 = \mathbf{0}$. Settings 1, 2 and 3 have been run 10 times, with different demand realizations generated through independent samples.

Figure 1 shows, for every setting, the regret obtained after $T$ periods as a function of the learning rate parameter $\gamma \in [10^{-5}, 10^1]$. We picked $T = 1969$ for all the settings because it corresponds to the number of periods available in our real-world dataset [17]. We see that MaxCOSD performs well compared to baselines whenever the number of handled products remains low (Settings 1,2,3,5). Instead, we see that MaxCOSD is less efficient when the number of products becomes large, in particular in the Setting 4. The performance of MaxCOSD in the large $n$ regime can be explained by the fact that the cycles become longer, as it becomes less likely that the feasibility condition is satisfied. Indeed, we have seen in Lemma 15 that the larger the overall demand is, the easier feasibility holds. But when $n$ grows, $\min_i d_{i,t}$ becomes smaller, making the problem harder.

## 7 Conclusion

In this paper, we address Online Inventory Optimization problems by introducing MaxCOSD, the first algorithm which can be applied to a wide variety of real-world problems. More precisely, MaxCOSD enjoys an optimal $O(\sqrt{T})$ regret without assuming the demand to be *i.i.d.*, and can handle a large class of dynamics, including perishability models. We achieved this result by applying ideas and methods from online learning to OIO problems.

Still, there is a lot of space for improvements and future developments. First, we observed that for problems with a large number of products and capacity constraints, the empirical performance of MaxCOSD could be improved. To do so, one would need to better handle the feasibility constraint, by using for instance projections onto the feasibilitly constraint, an idea already used by DDM, but with no theoretical guarantees so far in real-world scenarios. Second, we obtained regret bounds under minimal structural assumptions on the problem (convex lipschitz losses), but we could expect to obtain better rates by making stronger assumptions on the problem. One such assumption, which is classical in the Online Convex Optimization literature, is to assume further that the losses are strongly convex or exp-concave, typically leading to a logarithmic $O(\log(T))$ regret. Another assumption,

which is more specific to the Online Inventory literature, is to make some regularity assumption directly on the demand, also leading to a logarithmic pseudo-regret in the newsvendor case [3, Subsection 2.3]. Finally, even if our model is quite versatile, it has its limits, and more work is needed to improve it. For instance, we have seen in Section 2.2 that we do not accommodate for outdating costs as they appear in [32]. Also, we do not handle discrete feasible sets for which two promising techniques have been applied in the literature: expert algorithms [15] and probabilistic rounding [9, Subsection 3.4]. We could also hope to further weaken our non-degeneracy Assumption 10, by making an hypothesis which applies independently to each product, without having to assume pairwise independence across products. We believe that an adequate adaptation of online convex optimization techniques to the online inventory framework will prove to be a successful strategy for overcoming those challenges.

## Acknowledgement

We are thankful to Adeline Fermanian and the anonymous reviewers for helpful discussions and suggestions. Our work has been funded by the LOPF LabCom ANR-20-LCV1-0005 grant, provided by the French "Agence Nationale de la Recherche".

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

# A Analysis of MaxCOSD

The goal of this appendix is to study the MaxCOSD algorithm (see Algorithm 2) and prove Theorem 12. To do so, we start by introducing a generalized version of MaxCOSD named Cyclic Online Subgradient Descent (COSD) which has general update periods and learning rates (see Algorithm 3). In Proposition 19 we provide a general analysis of COSD which shows that it is sufficient to control the cycles' length to derive $O(\sqrt{T})$ regret bounds. As a byproduct of this proposition we also derive the classical analysis of OSD (see Corollary 20). A way of ensuring that the cycles' lengths are efficiently controlled is through a probabilistic property (see Assumption 21) which is similar to sub-exponential concentration [29, Section 2.7]. Finally, using Lemma 15 and Assumptions 2, 10 and 11 we show that when $\gamma \in (0, \rho/D]$ the cycles' length of MaxCOSD satisfy Assumption 21 which allows us to conclude with Theorem 12.

## A.1 Design of COSD

The Cyclic Online Subgradient Descent (COSD) generalizes MaxCOSD by allowing arbitrary learning rates and update periods. Recall that update periods are allowed to be dynamically defined, for this reason we will refer to a sequence of update periods $(t_k)_{k \in \mathbb{N}}$ as an *update strategy* which is formally defined below.

**Definition 18** (Update strategy). An update strategy $(t_k)_{k \in \mathbb{N}} \subset \mathbb{N}$ is a sequence of random variables such that, $t_1 = 1$, $t_k < t_{k+1}$ for every $k \in \mathbb{N}$, and for every $t \in \mathbb{N}$, $k \in \mathbb{N}$, the event $\{t_k = t\}$ is observable by the manager at the beginning of period $t$, *i.e.* it belongs to $\sigma(g_1, x_2, \ldots, g_{t-1}, x_t)$.

The pseudo-code of COSD is given in Algorithm 3.

---
**Algorithm 3: COSD**

---
1 **Parameters:** learning rates $(\eta_t)_{t \in \mathbb{N}}$, update strategy $(t_k)_{k \in \mathbb{N}}$ and $y_1 \in \mathcal{Y}$
2 **Initialization**
3      Set $t_1 = 1$, $k = 1$ ;
4 **for** $t = 1, 2, \ldots$ **do**
5      Output $y_t$;
6      Observe $g_t \in \partial \ell_t(y_t)$ and $x_{t+1}$;
7      **if** $t_{k+1} = t + 1$ **then**
8          Set $y_{t+1} = \text{Proj}_{\mathcal{Y}}\left(y_{t_k} - \eta_t \sum_{s=t_k}^{t} g_s\right)$;
9          Set $k = k + 1$;
10      **else**
11          Set $y_{t+1} = y_t$;

---

By choosing appropriately the update strategy we recover the following algorithms:

- **OSD.** If updates are made at every period, *i.e.* $t_k = k$, we recover OSD. Its feasibility is not guaranteed.

- **Minibatch OSD.** Given a fixed minibatch size $\tau \in \mathbb{N}$, the updates period are defined offline via $t_k = 1 + (k-1)\tau$.

- **CUP.** For every $k \in \mathbb{N}$, $t_k$ is defined as the periods where the inventory is empty, that is,

$$t_1 = 1, \qquad t_{k+1} = \inf\{t \geq t_k + 1 : x_t \preceq \mathbf{0}\} \text{ for } k \in \mathbb{N}. \tag{10}$$

  This update strategy corresponds to that used by CUP [32]. One easily sees that for such update periods $t_k$, feasibility always holds.

- **MaxCOSD.** To recover MaxCOSD we define dynamically the update strategy as the periods where the candidate order-up-to levels $(\hat{y}_t)_{t \in \mathbb{N}}$ are feasible. Formally, it writes:

$$t_1 = 1, \qquad t_{k+1} = \inf\{t \geq t_k + 1 : x_t \preceq \hat{y}_t\} \text{ for } k \in \mathbb{N}, \tag{11}$$

where $(\hat{y}_t)_{t\in\mathbb{N}}$ is defined by:

$$\hat{y}_1 = y_1, \quad \hat{y}_{t+1} = \text{Proj}_{\mathcal{Y}}\left(\hat{y}_{t_k} - \eta_t \sum_{s=t_k}^{t} g_s\right), \tag{12}$$

with $k = \max\{j \geq 1 : t_j \leq t\}$ or equivalently $t \in \mathcal{T}_k = \{t_k, \ldots, t_{k+1} - 1\}$. Notice that at update periods the implemented order-up-to levels coincide with the candidate order-up-to level, that is, we have $y_{t_k} = \hat{y}_{t_k}$ for all $k \in \mathbb{N}$.

## A.2 Analysis of COSD and OSD

The following proposition summarizes the main properties of COSD. For convenience, we will use the notation $\bar{t}_k := t_{k+1} - 1$ for the last period of the $k^{th}$ cycle $\mathcal{T}_k$.

**Proposition 19.** *Let assumptions 2.i) and 2.ii) be satisfied. Given any update strategy and any sequence of learning rates $(\eta_t)_{t\in\mathbb{N}}$ such that: $0 < \eta_{\bar{t}_{k+1}} \leq \eta_{\bar{t}_k}$ for all $k \in \mathbb{N}$, COSD (see Algorithm 3) has the following properties:*

*i)  it is feasible if and only if for all $k \in \{2, 3, \ldots\}$, $y_{t_k} \succeq x_{t_k}$,*

*ii)  for all $K \in \mathbb{N}$, the regret at the end of the $K^{th}$ cycle satisfies,*

$$R_{\bar{t}_K} \leq \frac{D^2}{2\eta_{\bar{t}_K}} + \frac{1}{2}\sum_{k=1}^{K} \eta_{\bar{t}_k} \left\|\sum_{t\in\mathcal{T}_k} g_t\right\|_2^2. \tag{13}$$

*Proof.* Let us prove claim i). By definition, the algorithm produces feasible sequence of order-up-to levels if $y_t \succeq x_t$ for all $t \in \mathbb{N}$. But for all $t \in \{t_k + 1, \ldots, \bar{t}_k\}$ for some $k \in \mathbb{N}$, we have necessarily $y_t = y_{t-1} \succeq [y_{t-1} - d_{t-1}]^+ \succeq x_t$ where the last inequality comes from (2). Thus we only need to check feasibility at the update periods, *i.e.* check that $y_{t_k} \succeq x_{t_k}$ for all $k \in \mathbb{N}$. Furthermore, it is clear that the latter is verified for $k = 1$ since $x_{t_1} = x_1 = \mathbf{0} \preceq y_{t_1}$.

Now to prove the regret bound (claim ii)) we follow the lines of the classical analysis of OSD (see e.g. the proof of [8, Theorem 3.1] or that of [21, Theorem 2.13]) when run against the sequence of losses $(\sum_{t\in\mathcal{T}_k} \ell_t)_{k\in\mathbb{N}}$ instead of $(\ell_t)_{t\in\mathbb{N}}$. For convenience, we will write $\tilde{g}_k = \sum_{t\in\mathcal{T}_k} g_t$ for all $k \in \mathbb{N}$.

Let $y \in \mathcal{Y}, K \in \mathbb{N}$ and $k \in [K]$, we start by bounding $\sum_{t\in\mathcal{T}_k} \ell_t(y_t) - \ell_t(y)$. By definition of the subgradient $g_t \in \partial \ell_t(y_t)$ we have:

$$\sum_{t\in\mathcal{T}_k} \ell_t(y_t) - \ell_t(y) = \sum_{t\in\mathcal{T}_k} \ell_t(y_{t_k}) - \ell_t(y) \leq \langle \tilde{g}_k, y_{t_k} - y\rangle. \tag{14}$$

We rewrite this bound as follows:

$$\langle \tilde{g}_k, y_{t_k} - y\rangle = \frac{1}{2\eta_{\bar{t}_k}}\left(\|y_{t_k} - y\|_2^2 + \eta_{\bar{t}_k}^2 \|\tilde{g}_k\|_2^2 - \|(y_{t_k} - y) - \eta_{\bar{t}_k}\tilde{g}_k\|_2^2\right).$$

By using the property of non-expansiveness of the Euclidean projection we have:

$$\left\|y_{t_{k+1}} - y\right\|_2^2 = \left\|\text{Proj}_{\mathcal{Y}}\left(y_{t_k} - \eta_{\bar{t}_k}\tilde{g}_k\right) - \text{Proj}_{\mathcal{Y}}(y)\right\|_2^2 \leq \left\|(y_{t_k} - \eta_{\bar{t}_k}\tilde{g}_k) - y\right\|_2^2.$$

Combining the last two results leads to:

$$\langle \tilde{g}_k, y_{t_k} - y\rangle \leq \frac{1}{2\eta_{\bar{t}_k}}\left(\|y_{t_k} - y\|_2^2 + \eta_{\bar{t}_k}^2 \|\tilde{g}_k\|_2^2 - \left\|y_{t_{k+1}} - y\right\|_2^2\right).$$

Finally, we combine the last inequality with inequality (14) and sum these over $k = 1, \ldots, K$.

$$
\begin{aligned}
\sum_{t=1}^{\bar{t}_K} \ell_t(y_t) - \ell_t(y) &\leq \sum_{k=1}^{K} \langle \tilde{g}_k, y_{t_k} - y \rangle \\
&\leq \sum_{k=1}^{K} \frac{1}{2\eta_{\bar{t}_k}} \left( \|y_{t_k} - y\|_2^2 - \|y_{t_{k+1}} - y\|_2^2 \right) + \sum_{k=1}^{K} \frac{\eta_{\bar{t}_k}}{2} \|\tilde{g}_k\|_2^2 \\
&= \frac{1}{2} \left( \frac{\|y_{t_1} - y\|_2^2}{\eta_{\bar{t}_1}} - \frac{\|y_{t_{K+1}} - y\|_2^2}{\eta_{\bar{t}_K}} + \sum_{k=1}^{K-1} \left( \frac{1}{\eta_{\bar{t}_{k+1}}} - \frac{1}{\eta_{\bar{t}_k}} \right) \|y_{t_{k+1}} - y\|_2^2 \right) \\
&\quad + \sum_{k=1}^{K} \frac{\eta_{\bar{t}_k}}{2} \|\tilde{g}_k\|_2^2 \\
&\leq \frac{1}{2} \left( \frac{D^2}{\eta_{\bar{t}_1}} + \sum_{k=1}^{K-1} \left( \frac{1}{\eta_{\bar{t}_{k+1}}} - \frac{1}{\eta_{\bar{t}_k}} \right) D^2 \right) + \sum_{k=1}^{K} \frac{\eta_{\bar{t}_k}}{2} \|\tilde{g}_k\|_2^2 \\
&= \frac{D^2}{2\eta_{\bar{t}_K}} + \sum_{k=1}^{K} \frac{\eta_{\bar{t}_k}}{2} \|\tilde{g}_k\|_2^2 .
\end{aligned}
$$

Claim ii) is thereby proved by taking the supremum over $y \in \mathcal{Y}$. $\qquad \square$

Proposition 19 gives us guidelines to design optimal update cycles and learning rates. First, this proposition states that it suffices to verify the feasibility constraint at the update periods to ensure that the whole sequence of order-up-to levels produced by COSD is feasible. On the other hand, to guarantee that this algorithm has a sublinear regret we will need to ensure that the cycles $\mathcal{T}_k$ are not too long, *i.e.* that update periods are frequent enough, and then consider adequate learning rates.

Since OSD is an instance of COSD, we can recover classical regret bounds of OSD (see e.g. [8, Theorem 3.1] or [21, Theorem 2.13]) and in particular $O(\sqrt{T})$ regret bounds, as a corollary of Proposition 19.

**Corollary 20.** *Let assumptions 2.i) and 2.ii) be satisfied. Given positive non-increasing learning rates $(\eta_t)_{t \in \mathbb{N}}$, i.e. $0 < \eta_{t+1} \leq \eta_t$ for all $t \in \mathbb{N}$, the regret of OSD (see Algorithm 1) satisfies for all $T \in \mathbb{N}$,*

$$
R_T \leq \frac{D^2}{2\eta_T} + \sum_{t=1}^{T} \frac{\eta_t}{2} \|g_t\|_2^2 .
$$

*In particular, if $\eta_t = \gamma D / (G\sqrt{t})$ with $\gamma > 0$, we have*

$$
R_T \leq \left( \frac{1}{2\gamma} + \gamma \right) GD\sqrt{T}.
$$

*Proof.* The regret bound follows from claim ii) of Proposition 19. Indeed, by taking $t_k = k$ for all $k \in \mathbb{N}$, COSD coincide with OSD, thus, for $K = \bar{t}_K = T$ we obtain the desired regret bound. $\quad \square$

## A.3 Controlling the cycles' length

In this subsection we analyze COSD with eventually unbounded cycles as it is the case for the CUP or the MaxCOSD update strategy. We start by introducing an assumption on the cycles called *geometric cycles* which yields expected regret bounds and high probability regret bounds.

**Assumption 21** (Geometric cycles). Let $\mu \in (0, 1]$. An update strategy $(t_k)_{k \in \mathbb{N}}$ has $\mu-$*geometric cycles* if there exists $C_\mu \geq 1$ such that for any $m \in \mathbb{N}$ and $k \in \mathbb{N}$ we have:

$$
\mathbb{P}\left[ t_{k+1} - t_k > m \right] \leq C_\mu (1 - \mu)^m. \tag{15}
$$

The name geometric cycles is motivated by the fact that if $\xi$ is a geometric random variable with parameter $\mu$, *i.e.* $\xi \in \mathbb{N}$ and $\mathbb{P}[\xi > m] = (1 - \mu)^m$ for all $m \in \mathbb{N}$, then Assumption 21 rewrites:

$\mathbb{P}[t_{k+1} - t_k > m] \leq C_\mu \mathbb{P}[\xi > m]$ for all $k, m \in \mathbb{N}$. This property resembles sub-exponential concentration [29, Section 2.7]. This notion was inspired from a proof in [32] where the authors observe, under their assumptions, that CUP's cycles length are geometrically distributed with a parameter lower bounded by $\mu$. Assumption 21 generalizes this property.

The following proposition summarizes some important properties of geometric cycles.

**Proposition 22.** *Consider an update strategy* $(t_k)_{k \in \mathbb{N}}$ *with* $\mu-$*geometric cycles (see Assumption 21), then,*

- *For all* $k \in \mathbb{N}$, $\mathbb{E}[t_{k+1} - t_k] \leq C_\mu/\mu$ *and* $\mathbb{E}\left[(t_{k+1} - t_k)^2\right] \leq C_\mu(2 - \mu)/\mu^2 \leq 2C_\mu/\mu^2$.

- *For all* $K \in \mathbb{N}$ *and any confidence level* $\delta \in (0, 1)$ *we have with probability at least* $1 - \delta$,

$$\sqrt{\sum_{k=1}^{K}(t_{k+1} - t_k)^2} \leq \left(1 + \frac{\log(KC_\mu/\delta)}{\mu}\right)\sqrt{K}.$$

*Proof.* First, let us observe that if $\mu = 1$ in Assumption 21 then, $t_k = k$ for all $k \in \mathbb{N}$ and one can easily check that all the claims are indeed verified. In the following we assume $\mu \in (0, 1)$.

Let $\xi$ be a geometric random variable of parameter $\mu$, that is, $\xi \in \mathbb{N}$ and $\mathbb{P}[\xi > m] = (1 - \mu)^m$ for all $m \in \mathbb{N}$. Then, it is well-known that $\mathbb{E}[\xi] = 1/\mu$ and $\mathbb{E}[\xi^2] = (2 - \mu)/\mu^2$.

We now prove the first claim by means of direct computations. For all $k \in \mathbb{N}$, we have:

$$\mathbb{E}[t_{k+1} - t_k] = \sum_{m=0}^{+\infty} \mathbb{P}[t_{k+1} - t_k > m] \leq \sum_{m=0}^{+\infty} C_\mu \mathbb{P}[\xi > m] = C_\mu \mathbb{E}[\xi] = \frac{C_\mu}{\mu}.$$

Also, using the fact that for any integer $a$ and real number $b$, we have, $a > b$ if and only if $a > \lfloor b \rfloor$, we can upper bound $\mathbb{E}\left[(t_{k+1} - t_k)^2\right]$ as follows:

$$\mathbb{E}\left[(t_{k+1} - t_k)^2\right] = \sum_{m=0}^{+\infty} \mathbb{P}\left[(t_{k+1} - t_k) > \sqrt{m}\right] = \sum_{m=0}^{+\infty} \mathbb{P}\left[(t_{k+1} - t_k) > \lfloor\sqrt{m}\rfloor\right]$$

$$\leq \sum_{m=0}^{+\infty} C_\mu \mathbb{P}\left[\xi > \lfloor\sqrt{m}\rfloor\right] = C_\mu \mathbb{E}[\xi^2] = C_\mu \frac{2 - \mu}{\mu^2} \leq \frac{2C_\mu}{\mu^2}.$$

Finally, let us prove the second claim. Let $K \in \mathbb{N}$ and $\varepsilon > 0$, it is classical to see that:

$$\mathbb{P}\left[\sqrt{\sum_{k=1}^{K}(t_{k+1} - t_k)^2} > \varepsilon\right] \leq \mathbb{P}\left[\exists k \in [K], \ (t_{k+1} - t_k)^2 > \varepsilon^2/K\right]$$

$$\leq \sum_{k=1}^{K} \mathbb{P}\left[(t_{k+1} - t_k) > \lfloor\varepsilon/\sqrt{K}\rfloor\right].$$

We use Assumption 21, which yields:

$$\mathbb{P}\left[\sqrt{\sum_{k=1}^{K}(t_{k+1} - t_k)^2} > \varepsilon\right] \leq KC_\mu(1 - \mu)^{\lfloor\varepsilon/\sqrt{K}\rfloor} \leq KC_\mu(1 - \mu)^{(\varepsilon/\sqrt{K})-1}.$$

Now for $\delta \in (0, 1)$ we plug $\varepsilon = \left(1 + \frac{\log(\delta/(KC_\mu))}{\log(1-\mu)}\right)\sqrt{K} > 0$ in the last result, to obtain:

$$\sqrt{\sum_{k=1}^{K}(t_{k+1} - t_k)^2} \leq \left(1 + \frac{\log(\delta/(KC_\mu))}{\log(1 - \mu)}\right)\sqrt{K},$$

with probability at least $1 - \delta$. We conclude by simply observing that $-1/\log(1 - \mu) \leq 1/\mu$.  $\square$

Using these tools, we are now ready to provide strong regret bounds for COSD under the assumption of geometric cycles. We are going from now on to focus on adaptive learning rates (see Eq. 7) which will allow us to unlock high probability regret bounds.

**Corollary 23.** *Consider an inventory problem satisfying Assumption 2 and COSD (see Algorithm 3) with adaptive learning rates (see Eq. (7)) and an update strategy with $\mu-$geometric cycles (see Assumption 21), then, the following regret bounds hold for all $T \in \mathbb{N}$,*

$$\mathbb{E}\left[R_T\right] \leq \frac{\sqrt{2C_\mu}}{\mu}DG\left(\frac{1}{2\gamma}+\gamma+1\right)\sqrt{T},$$

*and for any confidence level $\delta \in (0,1)$ we have with probability at least $1-\delta$,*

$$R_T \leq DG\left(\frac{1}{2\gamma}+\gamma+1\right)\left(1+\frac{1}{\mu}\log\left(\frac{TC_\mu}{\delta}\right)\right)\sqrt{T}.$$

*Finally, COSD is feasible if and only if for all $k \in \{2,3,\dots\}$, $y_{t_k} \succeq x_{t_k}$.*

*Proof.* Let $T \in \mathbb{N}$ and $K = \min\{k \geq 1, t_k \leq T\}$. We start by bounding the regret $R_T$ in terms of the regret at the end of $K-1^{th}$ cycle $R_{\bar{t}_{K-1}} = R_{t_K-1}$ and a remainder term as follows:

$$R_T \leq R_{\bar{t}_{K-1}} + \sup_{y \in \mathcal{Y}}\sum_{t=t_K}^{T}\ell_t(y_t)-\ell_t(y) \leq R_{\bar{t}_{K-1}} + \sup_{y \in \mathcal{Y}}\sum_{t=t_K}^{T}\langle g_t, y_t-y\rangle.$$

where we used $g_t \in \partial\ell_t(y_t)$. Applying Cauchy-Schwartz inequality, Assumption 2 and $T \leq t_{K+1}-1$ leads us to:

$$R_T \leq R_{\bar{t}_{K-1}} + DG(t_{K+1}-t_K). \tag{16}$$

Let us now bound $R_{\bar{t}_{K-1}}$. As a consequence of claim ii) of Proposition 19 and after substituting $\eta_t$ by its value and applying Lemma 25 provided in Appendix B to $f(x) = 1/(2\sqrt{x})$, $a_0 = 0$ and $a_k = \left\|\sum_{t\in\mathcal{T}_k}g_t\right\|_2^2$ for $k \in [K-1]$ we obtain:

$$R_{\bar{t}_{K-1}} \leq D\left(\frac{1}{2\gamma}+\gamma\right)\sqrt{\sum_{k=1}^{K-1}\left\|\sum_{t\in\mathcal{T}_k}g_t\right\|_2^2} \leq DG\left(\frac{1}{2\gamma}+\gamma\right)\sqrt{\sum_{k=1}^{K-1}(t_{k+1}-t_k)^2}.$$

Combining this with inequality (16) leads us to:

$$R_T \leq DG\left(\left(\frac{1}{2\gamma}+\gamma\right)\sqrt{\sum_{k=1}^{K-1}(t_{k+1}-t_k)^2}+(t_{K+1}-t_K)\right)$$

$$\leq DG\left(\frac{1}{2\gamma}+\gamma+1\right)\sqrt{\sum_{k=1}^{K}(t_{k+1}-t_k)^2}.$$

To obtain the expected regret bound, we start by noticing that $K \leq T$ then taking the expectation in this last inequality, applying Jensen's inequality, then, Proposition 22 that bounds $\mathbb{E}\left[(t_{k+1}-t_k)^2\right] \leq 2C_\mu/\mu^2$ we end up with the desired bound. Finally, to obtain the high probability regret bound, we notice again that $K \leq T$ and then apply the high probability bound of Proposition 22. $\square$

### A.4 Proof of Theorem 12

In the following lemma we claim that under the assumptions of Theorem 12, MaxCOSD with the appropriate learning rates has geometric cycles (see Assumption 21).

**Lemma 24.** *Consider an inventory problem and let assumptions 2, 10 and 11 hold. Then, MaxCOSD with learning rates defined in Eq. (7) and $\gamma \in (0, \rho/D]$ has $\mu-$geometric cycles with $C_\mu = 1$.*

*Proof.* First, notice that if $\min_{i\in[n]}d_{t,i} \geq \rho$ then $x_{t+1} \preceq \hat{y}_{t+1}$, i.e. $t+1$ is an update period for MaxCOSD. This is a consequence of Lemma 15, which applies since we have:

$$\|\hat{y}_{t+1}-y_t\| = \left\|\text{Proj}_{\mathcal{Y}}\left(\hat{y}_{t_k}-\eta_t\sum_{s=t_k}^{t}g_s\right)-\hat{y}_{t_k}\right\| \leq \eta_t\left\|\sum_{s=t_k}^{t}g_s\right\|_2 \leq \gamma D \leq \rho \leq \min_{i\in[n]}d_{t,i}.$$

Now let $k \in \mathbb{N}$ and $m \in \mathbb{N}$. Using this initial observation we have:

$$\mathbb{P}\left[t_{k+1} - t_k > m\right] = \mathbb{P}\left[x_{t_k+1} \not\preceq \hat{y}_{t_k+1}, \ldots, x_{t_k+m} \not\preceq \hat{y}_{t_k+m}\right]$$

$$\leq \mathbb{P}\left[\min_{i \in [n]} d_{t_k,i} < \rho, \ldots, \min_{i \in [n]} d_{t_k+m-1,i} < \rho\right]$$

$$= \sum_{s \geq 1} \mathbb{P}\left[t_k = s, \min_{i \in [n]} d_{s,i} < \rho, \ldots, \min_{i \in [n]} d_{s+m-1,i} < \rho\right]$$

The last step of this proof is showing that in fact,

$$\mathbb{P}\left[t_k = s, \min_{i \in [n]} d_{s,i} < \rho, \ldots, \min_{i \in [n]} d_{s+m-1,i} < \rho\right] \leq \mathbb{P}\left[t_k = s\right](1-\mu)^m. \tag{17}$$

We prove this by a simple induction over $m \geq 1$. Noticing that $\{t_k = s\} \in \sigma(g_1, x_2, \ldots, g_{s-1}, x_s) \subset \sigma(\ell_1, d_1, \ldots, \ell_{s-1}, d_{s-1})$, where the last inclusion comes from Assumption 11, and using the basic properties of conditional expectations we derive the inequality for $m = 1$:

$$\mathbb{P}\left[t_k = s, \min_{i \in [n]} d_{s,i} < \rho\right] = \mathbb{E}\left[\mathbb{P}\left[t_k = s, \min_{i \in [n]} d_{s,i} < \rho | \ell_1, d_1, \ldots, \ell_{s-1}, d_{s-1}\right]\right]$$

$$= \mathbb{E}\left[\mathbb{1}_{\{t_k=s\}} \mathbb{P}\left[\min_{i \in [n]} d_{s,i} < \rho | \ell_1, d_1, \ldots, \ell_{s-1}, d_{s-1}\right]\right]$$

$$\leq \mathbb{P}\left[t_k = s\right](1-\mu),$$

where the last inequality comes from Assumption 10. Assume now the relation (17) holds for $m$, let us prove in a similar way that it holds for $m + 1$.

$$\mathbb{P}\left[t_k = s, \min_{i \in [n]} d_{s,i} < \rho, \ldots, \min_{i \in [n]} d_{s+m,i} < \rho\right]$$

$$= \mathbb{E}\left[\mathbb{P}\left[t_k = s, \min_{i \in [n]} d_{s,i} < \rho, \ldots, \min_{i \in [n]} d_{s+m,i} < \rho | \ell_1, d_1, \ldots, \ell_{s+m-1}, d_{s+m-1}\right]\right]$$

$$= \mathbb{E}\left[\mathbb{1}_{\{t_k=s\}} \mathbb{1}_{\{\min_{i \in [n]} d_{s,i} < \rho\}} \cdots \mathbb{1}_{\{\min_{i \in [n]} d_{s+m-1,i} < \rho\}} \mathbb{P}\left[\min_{i \in [n]} d_{s+m,i} < \rho | \ell_1, d_1, \ldots, \ell_{s+m-1}, d_{s+m-1}\right]\right]$$

$$\leq \mathbb{P}\left[t_k = s, \min_{i \in [n]} d_{s,i} < \rho, \ldots, \min_{i \in [n]} d_{s+m-1,i} < \rho\right](1-\mu)$$

$$\leq \mathbb{P}\left[t_k = s\right](1-\mu)^{m+1}.$$

Summing the relations (17) over $s \geq 1$ leads to the final bound: $\mathbb{P}\left[t_{k+1} - t_k > m\right] \leq (1-\mu)^m$, which is our claim. □

We are now ready to prove Theorem 12.

*Proof of Theorem 12.* By definition, MaxCOSD is always feasible (independently of the learning rates chosen). Lemma 24 ensures that when $\gamma \in (0, \rho/D]$, MaxCOSD with adaptive learning rates as defined in Eq. (7) has $\mu-$geometric cycles with $C_\mu = 1$. Thus, Corollary 23 applies and leads to the regret bounds we claimed. □

## B  Other lemmas

**Lemma 25** (Lemma 4.13 of [21]). *Let $a_0, a_1, \ldots, a_K$ be non-negative numbers and $f : \mathbb{R}_+ \to \mathbb{R}_+$ a measurable non-increasing function, we have:*

$$\sum_{k=1}^{K} a_k f\left(a_0 + \sum_{m=1}^{k} a_m\right) \leq \int_{a_0}^{\sum_{k=0}^{K} a_k} f(x) dx.$$

*Proof.* Define $s_k = \sum_{m=0}^{k} a_m$. The following holds for all $k \in [K]$,

$$a_k f \left( a_0 + \sum_{m=1}^{k} a_m \right) = a_k f(s_k) = \int_{s_{k-1}}^{s_k} f(s_k) dx \leq \int_{s_{k-1}}^{s_k} f(x) dx.$$

Summing over $k = 1, \ldots, K$ leads to the desired bound. $\square$

**Lemma 26.** *Consider an inventory problem with non-negative feasible set $\mathcal{Y} \subset \mathbb{R}_+^n$. Then, any non-decreasing sequence of order-up-to levels $(y_t)_{t \in \mathbb{N}} \subset \mathcal{Y}$ (i.e. such that $y_t \preceq y_{t+1}$ for all $t \in \mathbb{N}$) is feasible. In particular, constant strategies are feasible too.*

*Proof.* Consider a non-decreasing sequence of order-up-to levels $(y_t)_{t \in \mathbb{N}} \subset \mathcal{Y} \subset \mathbb{R}_+^n$. First of all, $y_1$ is feasible since we impose $x_1 = \mathbf{0}$ and $y_1 \in \mathbb{R}_+^n$. Then, for every $t \in \mathbb{N}$, we can write:

$$y_{t+1} \succeq y_t \succeq [y_t - d_t]^+ \succeq x_{t+1}.$$

The first inequality comes from the assumption that $(y_t)_{t \in \mathbb{N}}$ is non-decreasing, the second one from the monotonicity of the positive part and the fact that $y_t \in \mathbb{R}_+^n$, finally, the last inequality is the inventory dynamical constraint. Thus, the whole strategy $(y_t)_{t \in \mathbb{N}}$ is feasible. $\square$

**Lemma 27.** *Consider an inventory problem that satisfies Assumption 2, then, the regret of any algorithm is bounded as follows:*

$$R_T \leq DGT.$$

*Proof.* Let $y \in \mathcal{Y}$. For all $t \in \mathbb{N}$, we have:

$$\ell_t(y_t) - \ell_t(y) \leq \langle g_t, y_t - y \rangle \leq \|g_t\|_2 \|y_t - y\|_2 \leq GD,$$

where we used the definition of the subgradient $g_t \in \partial \ell_t(y_t)$, then, Cauchy-Schwartz inequality and finally Assumption 2. Summing these inequalities over $t = 1, \ldots, T$ and taking the supremum over $y \in \mathcal{Y}$ leads to the desired bound. $\square$

**Lemma 28.** *Consider the newsvendor cost function $c$ defined in (5). Let $d \in \mathbb{R}^n$. A vector $g \in \mathbb{R}^n$ is a subgradient of the function of $c(\cdot, d)$ at $y \in \mathbb{R}^n$ if and only if for all $i \in [n]$ we have:*

$$g_i \in \begin{cases} \{h_i\}, & \text{if } y_i > d_i \\ [-p_i, h_i], & \text{if } y_i = d_i \\ \{-p_i\}, & \text{if } y_i < d_i. \end{cases}$$

*In particular, denoting $s = \min\{y, d\}$, the vector $(h_i \mathbb{1}_{\{y_i > s_i\}} - p_i \mathbb{1}_{\{y_i = s_i\}})_{i \in [n]}$ is a subgradient of $c(\cdot, d)$ at $y$.*

## C  Postponed proofs

### C.1  Proof of Proposition 13

*Proof of Proposition 13.* Formally, in the lost sales single-product newsvendor setting with observable demand over $\mathcal{Y} = [0, D]$, a feasible deterministic algorithm is defined by a sequence of functions $(Y_t)_{t \in \mathbb{N}}$ of the form $Y_t : \mathbb{R}_+^{t-1} \to [0, D]$ satisfying $Y_{t+1}(d_1, \ldots, d_t) \geq [Y_t(d_1, \ldots, d_{t-1}) - d_t]^+$ for all $d_1, \ldots, d_t \in \mathbb{R}_+$.

Let $\tilde{d} \in (0, D]$, and consider a constant demand sequence defined by $\tilde{d}_t := \tilde{d}$ at every period $t \in \mathbb{N}$. Consider now $(\tilde{y}_t)_{t \in \mathbb{N}}$ the sequence of order-up-to levels generated by this algorithm when facing this constant demand, that is, $\tilde{y}_t := Y_t(\tilde{d}, \ldots, \tilde{d})$. We will now distinguish two cases.

i) Consider the case where $\tilde{y}_t = 0$ for all $t \in \mathbb{N}$. Taking $y = \tilde{d}$ in the regret definition (3) yields:

$$R_T \geq \sum_{t=1}^{T} c(0, \tilde{d}) - \sum_{t=1}^{T} c(\tilde{d}, \tilde{d}) = Tp\tilde{d} = \Omega(T).$$

ii) Now, consider the case where there exists $T_0 \geq 1$ such that $\tilde{y}_1 = \cdots = \tilde{y}_{T_0-1} = 0$ and $\tilde{y}_{T_0} > 0$. Consider a new demand sequence $(d_t)_{t\in\mathbb{N}}$ defined as follows: $d_1 = \cdots = d_{T_0-1} = \tilde{d}$ and $d_t = 0$ for $t \geq T_0$. Denote by $(y_t)_{t\in\mathbb{N}}$ the sequence of order-up-to levels generated by the algorithm against the demand sequence $(d_t)_{t\in\mathbb{N}}$, that is, $y_t = Y_t(d_1, \ldots, d_{t-1})$.

Since the algorithm is deterministic, we also have: $y_t = 0$ for $t \leq T_0 - 1$. Indeed, we have, $y_t = Y_t(d_1, \ldots, d_{t-1}) = Y_t(\tilde{d}, \ldots, \tilde{d}) = \tilde{y}_t = 0$. On the other hand, we have for all $t \geq T_0+1$, $y_t \geq [y_{t-1} - d_{t-1}]^+ = y_{t-1}$, thus, $y_t \geq y_{T_0} = \tilde{y}_{T_0} > 0$ for all $t \geq T_0$.

Taking $y = 0$ in the regret definition (3) we obtain for $T \geq T_0$,

$$R_T \geq \sum_{t=1}^{T} c(y_t, d_t) - c(0, d_t) = \sum_{t=T_0}^{T} c(y_t, 0) = \sum_{t=T_0}^{T} h y_t \geq (T - T_0 + 1) h \tilde{y}_{T_0} = \Omega(T).$$

$\square$

### C.2 Proof of Proposition 14

*Proof of Proposition 14.* Take $(d_t)_{t\in\mathbb{N}} \subset \mathbb{R}_{++}$ such that $\sum_{t=1}^{\infty} d_t < y_1$ and define $C = y_1 - \sum_{t=1}^{\infty} d_t > 0$. Notice that due to the feasibility constraint (1) and the lost sales dynamic we have for all $t \in \mathbb{N}$, $y_{t+1} \geq x_{t+1} = [y_t - d_t]^+ \geq y_t - d_t$. Thus, we have for every $t \in \mathbb{N}$, $y_t = y_1 + \sum_{s=1}^{t-1}(y_{s+1} - y_s) \geq y_1 - \sum_{s=1}^{t-1} d_s \geq C$. Now take the losses $\ell_t(y) = y$, then, $R_T = \sum_{t=1}^{T} y_t \geq CT$ for all $T \in \mathbb{N}$. $\square$

### C.3 Proof of Lemma 15

*Proof of Lemma 15.* Since $[y - d]^+ = \max\{y-d, 0\}$ and $y' \succeq 0$, it is enough to show that $y' \succeq y-d$. The latter holds since, for any $i \in [n]$, we have $y_i - y'_i \leq \|y' - y\|_2 \leq \min_{i\in[n]} d_i \leq d_i$. $\square$

### C.4 Proof of Theorem 17

*Proof of Theorem 17.* The regret bound follows from the classical analysis of OSD, see our Corollary 20 or the proof of [8, Theorem 3.1]. Thus, we only need to show the feasibility. For all $t \in \mathbb{N}$, we have,

$$\|y_{t+1} - y_t\|_2 = \left\|\mathrm{Proj}_{\mathcal{Y}}(y_t - \eta_t g_t) - y_t\right\|_2 \leq \eta_t \|g_t\|_2 = \frac{\gamma D}{G\sqrt{t}} \|g_t\|_2 \leq \frac{\rho}{\sqrt{t}} \leq \rho, \quad (18)$$

the first inequality is provided by the property of non-expansiveness of the Euclidean projection, the second inequality comes from $\gamma \leq \rho/D$ and $\|g_t\|_2 \leq G$, and the last inequality from $\sqrt{t} \geq 1$. Combining this result with Assumption 16, we obtain $\|y_{t+1} - y_t\|_2 \leq \min_{i\in[n]} d_{t,i}$. According to Lemma 15 and the inventory dynamical constraint (2) this guarantees feasibility. $\square$

## D Discussion

### D.1 On the relation between Online Inventory Optimization and Online Convex Optimization

In the usual Online Convex Optimization (OCO) framework introduced by [36] a decision-maker and an environment interact as follows: at every time period $t \in \mathbb{N}$, first, the decision-maker chooses a decision $y_t \in \mathcal{Y}$ and the environment chooses a loss function $\ell_t$, then, the decision-maker receive some feedback which usually consist of either the loss function itself $\ell_t$ (full-information setting), the loss incurred $\ell_t(y_t)$ (bandit setting) or a subgradient $g_t \in \partial\ell_t(y_t)$ (first-order feedback setting). The goal of the decision-maker is to minimize its cumulative loss incurred $\sum_{t=1}^{T} \ell_t(y_t)$.

OIO extend the OCO framework by adding the feasibility constraint (1). A naive solution to accommodate such constraints into the OCO framework is by adding to the losses the convex indicator of the feasibility constraint, that is, by considering the losses $\tilde{\ell}_t(y) = \ell_t(y) + \chi_t(y)$, where $\chi_t(y)$ takes the value 0 if $y \succeq x_t$ holds and $+\infty$ otherwise. However, this is not satisfactory since by

doing so we also alter the regret (3) by imposing on the competitor $y \in \mathcal{Y}$ the feasibility constraints associated to the algorithm.

Many extensions of the OCO framework have been developed over the years. Of particular interest, those which include constraints of different forms like:

- *OCO with long-term constraints* [16, 11], where $m$ convex constraints of the form $f_i(\cdot) \leq 0$ for $i \in [m]$ should be satisfied in the long run, that is, the goal is to minimize the cumulative loss while keeping low constraint violation $\sum_{t=1}^{T} f_i(y_t)$ for each $i \in [m]$.

- *OCO with long-term and time-varying constraints* [20, 14] which compared to the previous extension, considers time-varying convex constraints of the form $f_{t,i}(\cdot) \leq 0$ where $f_{t,i}$ is revealed at the *end* of time period $t$. Even as long-term constraints, this learning task is known to be unsolvable in general (see e.g. [28, Proposition 2.1] or [18, Proposition 4] for more precise statement), thus, restricted notions of regret have been considered in this context.

- *OCO with ramp constraints* [2], where at each time period $t \in \{2, 3, \dots\}$ the decision-maker should choose $y_t \in \mathcal{Y}$ such that $|y_{t,i} - y_{t-1,i}| \leq \kappa_i$ for all $i \in [n]$.

We argue that OIO problems are different from these extensions. Indeed, our feasibility constraints (1) are neither long-term constraint since we do not allow for violations, nor ramp constraints since the bounds are time-varying. Also, our task is further challenging since we aim at bounding the regret (3) based on a competitor $y \in \mathcal{Y}$ which does not suffer from the feasibility constraint (1).

### D.2 On the notion of pseudo-regret

There exists an alternative notion of regret we call the *pseudo-regret* $\bar{R}_T$ which is in fact more common in the literature of online inventory problems [9, 3, 25, 32]. It is defined as the difference between the expected cumulative loss of the algorithm and that of the best fixed constant strategy, that is, formally

$$\bar{R}_T = \mathbb{E}\left[\sum_{t=1}^{T} \ell_t(y_t)\right] - \inf_{y \in \mathcal{Y}} \mathbb{E}\left[\sum_{t=1}^{T} \ell_t(y)\right].$$

The difference between the expected regret $\mathbb{E}[R_T]$ and the pseudo-regret $\bar{R}_T$ is in the competitor $y$. In the former, the competitor is random and depends on the realization of the losses, whereas, in the latter the competitor is fixed and depends only on the distribution of the losses. Notice that we always have $\bar{R}_T \leq \mathbb{E}[R_T]$, thus, an upper bound obtained on the expected regret applies directly to the pseudo-regret.

