# OpenReview forum: "Online Inventory Problems: Beyond the i.i.d. Setting with Online Convex Optimization"
_NeurIPS.cc/2023/Conference — NeurIPS 2023 poster_

### Official Review · Reviewer_8ZLu · 2023-06-15

**Soundness:** 2 fair
**Presentation:** 3 good
**Contribution:** 2 fair
**Rating:** 4
**Confidence:** 3

**Summary:**

In this paper, the authors consider the classic online inventory control problem. While previous works mainly consider the case where the demand is stochastically drawn from a stationary distribution, this paper considers the case where the demand can be adversarial. In addition, the authors consider a more general framework Online Inventory Optimization (OIO), which includes many different setups in the inventory literature, i.e. censored demand / backlogged demand / perishable / non-perishable. Then, the authors proposed the algorithm MaxCOSD achieving $O(\sqrt{T})$ regret under the assumption on the lower boundedness of the demand (with constant probability). The algorithm shares similar ideas to the cyclical updates proposed in CUP [Zhang et al., 2018] but with a different definition on the cycle and an adaptive choice of learning rate. The algorithm keeps using the same base-stock level to collect enough gradient information until the time where the outcome from the (cumulative) gradient descent step gives a feasible strategy. The authors also discuss their demand assumptions and show that without lower bounded demand assumption, no deterministic algorithm can achieve sublinear regret. Experiments are also done to show the effectiveness of the algorithm MaxCOSD.

**Strengths:**

- The problem considered in this paper is interesting and has wide applications in real life.
- The proposed algorithm is clean and easy to follow. The proposed framework OIO is general and includes many different setups that are considered in the inventory literature.
- Experiments on both synthetic data and real-world data are also done for their proposed algorithm.

**Weaknesses:**

- One major concern is the novelty of the proposed algorithm. From the technical perspective, I think the analysis generally follows the classic analysis of online gradient descent with a self-confidence tuning of the learning rate in order to handle the issue raised by the scale of the gradient that may cumulate over rounds. Based on this, I think the $\sqrt{T}$ regret is not very surprising.
- Several claims made in this paper may need more explanations or clarifications.
    - On Assumption 10, it is unclear how to transform the assumption that $\mathbb{E}[d]\geq \rho$ assumed in AIM to this assumption with the same theoretical guarantees. Specifically, with the current Assumption 10, when $\mu$ and $\rho$ is $\Theta(T^{-\alpha})$ and $\Theta(T^{-\beta})$, the regret bound becomes $T^{\frac{1}{2}+\alpha+\beta}$. It would be better if the authors can compare the guarantees of both algorithms under the same assumptions.
    - The lower bound analysis is problematic or not rigorous. For proposition 13, in line 601, the authors argue that
$(T-T_0+1)hy_{T_0}=\Omega(T)$ , which does not hold when ${y}_{T_0}=o(T)$. Similarly, in Proposition 14, $y_1$ can also be $o(T)$ making the regret sublinear. A refined analysis on both propositions should be included.
- The experiment results do not show a better performance of MaxCOSD compared to other algorithms in Settings 1-4, even though some algorithms are not designed to handle the adversarial/non-stationary demand. This weakens the motivation of providing an algorithm that performs better in the adversarial environment. In addition, for Settings 1-3, showing the error bar or the std of the algorithm would be better since the algorithm actually guarantees high-probability regret bound.

**Questions:**

-  In line 244, I do not understand why Assumption 10 with $\rho=D$ matches the assumption 1 made by CUP in [Zhang et al., 2018]. In CUP, they are assuming the knowledge of the upper bound on the optimal inventory level.
- On the optimality on the parameters $\rho$ and $\mu$ in Theorem 12: I wonder whether there is lower bound on $\mu$ and $\rho$ for this problem? This may show further tightness of the obtained bound.

**Limitations:**

See Weakness and Questions sections.

---

> ### Author Rebuttal · Authors · 2023-08-07
>
> 1. *One major concern is the novelty of the proposed algorithm. From the technical perspective, I think the analysis generally follows the classic analysis of online gradient descent with a self-confidence tuning of the learning rate in order to handle the issue raised by the scale of the gradient that may cumulate over rounds.*
>
>    See our global response.
>
> 2. *Based on this, I think the regret $O(\sqrt{T})$ is not very surprising.*
>
>    We would like to highlight here that applying a classic analysis of OSD to inventory cannot lead to $O(\sqrt{T})$ rates. This is what we prove in Proposition 13. To get this rate we need to make assumptions which is totally unusual for online learning.
>
> 3. *Several claims made in this paper may need more explanations or clarifications [...] The lower bound analysis is problematic or not rigorous [...] A refined analysis on both propositions should be included.*
>
>    We firmly disagree with the reviewer.
>    Each of the four critics raised by the reviewer rely on the assumption that some constants ($\mu$, $\rho$, $y_{T_0}$, or $y_1$) could depend on $T$. It is hard for us to make sense of this, as it is stated in our assumptions and statements that those quantities are *constant* and independant of $T$. This is for instance very clear in Assumption 10.
>    We conjecture that there is a confusion here, and that the reviewer has in mind an Online Learning setting where the horizon is *finite*. In such setting there is a fixed finite horizon $T \in \mathbb{N}$, and the learning dynamic unfolds on time periods $t=1,...,T$; at the end of which we compute a regret $R_T$. There we could indeed have parameters depending on the horizon $T$ (this happens in the corresponding literature), in which case the raised critics would be relevant. We stress that this is not our setting (we work with an *infinite horizon*).
>    There is also the possibility that the above is incorrect, and that we are completely missing something. In this case we would need further explanations from the reviewer?
>
> 4. *The experiment results do not show a better performance of MaxCOSD compared to other algorithms*
>
>    This is correct, and stated in the paper ("*we see that MaxCOSD is less efficient when the number of products becomes large*").
>    Please note that the goal of this paper is not to improve on state-of-the-art performances in a specific setting. Instead, we provide a versatile algorithm able to tackle various (and new) settings while being theoretically grounded. We believe that wanting an algorithm which applies to a broader setting while at the same time beating the algorithms designed for solving a specific setting is irrealistic. This being said, we should not compromise on performance either, and we show that besides hard problems, our method performs comparatively to competitors.
>
> 5. *This weakens the motivation of providing an algorithm that performs better in the adversarial environment.*
>
>  We never make the claim that our algorithm performs better in an adversarial environment.
>
> 6. *showing the error bar or the std of the algorithm would be better*
>
> We considered doing this but chose not to do it for two reasons.
> The first reason is that we use both synthetic data with real data, for which how to compute the std is not clear. The second reason is that it did not provide meaningful information for synthetic data. Indeed, in Settings 1-2 the variance is relatively very low overall ; and in Settings 3 the variance becomes significative only in the regime when $\gamma$ is large (for both algorithms), which makes the plots look messy, while having little interest since it is the regime where the algorithms performs less well. For the sake of the conversation, we share in the global response our plots with the std displayed.
>
> 7. *In line 244, I do not understand why Assumption 10 with $\rho=D$ matches the assumption 1 made by CUP in [Zhang et al., 2018]. In CUP, they are assuming the knowledge of the upper bound on the optimal inventory level.*
>
>    First of all, we emphasize that we do not make the claim that our Assumption 10 implies the Assumption 1 for CUP. Instead we say that we recover "*an*" assumption. Allow us to explain this in detail. Our point of view on the CUP algorithm is that Assumption 1 can be decomposed in two assumptions which have very different purposes.
>
> - First, the authors assume that the (unconstrained) problem they want to solve is equivalent to a problem with the constraint $\mathcal{Y} = [0, \bar S]$ . To do so, they assume that the optimal policy $S^*$ is lower than $\bar S$. This assumption allows them to retrieve some compactness (which our paper has since we assume $\mathcal{Y}$ to be compact).
> - Second, they make a non-degeneracy assumption on the demand to obtain $\sqrt{T}$ rates : $\mathbb{P}[d_t \geq \bar S] >0$. It is this assumption which is equivalent to our Assumption 10 when taking $\mathcal{Y}=[0,D]$ and $\rho = D$.
>   This is the reason why we did not refer to Assumption 1 as a whole in this sentence, but to *an* assumption.
>   We recognize that this is imprecise, and could be ambigous. In view to remove any ambiguity and make all this more understandable for the reader, we propose to add a sentence in Remark 6 (about the feasible set $\mathcal{Y}$) explaining that in various papers (the CUP one, but also the AIM one) the authors first assume to know an upper bound $\bar S$ on the optimal policy $S^*$, and then work with $\mathcal{Y} = [0,\bar S]$. This way our sentence in line 244 would make more sense, after eventually adding a call back to Remark 6.
>
> 8. *On the optimality on the parameters $\rho$ and $\mu$ [...]* :
>
>    We address this in our global response (in short: we do not know, and as far as we know other competitors obtain the same kind of constants).

---

> > ### Comment · Reviewer_8ZLu · 2023-08-20
> >
> > Thanks for the authors' response. However, I am still not convinced on the significance of the obtained results.
> >
> > - 1. We would like to highlight here that applying a classic analysis of OSD to inventory cannot lead to $O(\sqrt{T})$ rates. This is what we prove in Proposition 13. To get this rate we need to make assumptions which is totally unusual for online learning.
> >
> > Yes but I was referring to the $O(\sqrt{T})$ results with additional assumptions made (e.g. lower bounded density for the per-round demand distribution), which can be obtained by similar online gradient descent analysis in [9]. Also the adversarial demand lower bound is also not hard since you may accumulate too many inventory for one round while not getting these inventories sold in later rounds because of the adversarial demand.
> >
> > - 2. Several claims made in this paper may need more explanations or clarifications [...] The lower bound analysis is problematic or not rigorous [...] A refined analysis on both propositions should be included.
> >
> > I think the authors misunderstood my point in the sense that $y_1$ is a decision variable decided by **the algorithm** instead of the problem instance. If the algorithm decides $y_1$ to be $o(T)$, then your lower bound does not hold.
> >
> > - 3. This weakens the motivation of providing an algorithm that performs better in the adversarial environment.
> >
> > I was thinking the paper tries to consider the non-i.i.d demand case and design an algorithm which performs good in this environment (not necessarily adversarial). However, the experiments show that the performance of the algorithm designed for the i.i.d case (DDM) performs even better than the one proposed in this paper in setting 4, which is definitely not an i.i.d environment. This weakens the motivation of using the algorithm proposed in this paper, or in other words, whether the existing algorithms can already achieve so under these environment.

---

> > > ### Author Response · Authors · 2023-08-21
> > > **Answer about the misunderstanding regarding constants possibly depending on time**
> > >
> > > We again insist that $y_1$ is a constant real number chosen in the interval (0,D], as stated in our Proposition 14. It is not a  function, and can by no means depend on T (which is a mute variable). As explained in our rebuttal, T is not a quantity fixed beforehand, because we are not in a fixed horizon setting.
> > >
> > > To illustrate this and further convince the reviewer, we provide below a simple technical explanation.
> > >
> > > Assume we fix some $T\in\mathbb N$ (say T=10) and set $y_1=T^{-1/2}$, then we could obtain in the proof of this proposition, $C=T^{-1/2}/2$ by choosing an appropriate sequence of demands $(d_t)$, this would indeed lead to $R_T\geq (\sqrt T)/ 2$.
> > > This does *not* contradict our claim, since this inequality is only valid for this **single value** of $T$. Consider now any $T’\in\mathbb N$ (a mute variable), then we obtain $R_{T’} \geq CT’ = T^{-1/2}T’/2$ which means that the regret grows linearly with time, which conforms to our claim. The same argument holds for all the other constant parameters.

---

> > > ### Author Response · Authors · 2023-08-21
> > > **Answer about the significance of the results**
> > >
> > > *"the results [...] can be obtained by similar online gradient descent analysis in [9]"*
> > >
> > > We disagree with the claim reducing the results in [9] to a classic analysis of OGD. For instance, the key part of the proof in [9] relies on additional results from queuing theory to prove these rates [9, Paragraph 3.2.3], see also the proof of [9, Proposition 4] in [9, Appendix B].
> > > The final $O(\sqrt T)$ regret bound is indeed not very surprising but not so easy to obtain either. We recall that our work provide such rates for a significantly large range of settings, some of which have never been considered in the literature such as stateful non-iid settings or perishable multi-product settings.
> > >
> > >
> > > *"Also the adversarial demand lower bound is also not hard since you may accumulate too many inventory for one round while not getting these inventories sold in later rounds because of the adversarial demand"*
> > >
> > > Please note that we never claim that this lower bound is hard to obtain. We do claim this bound is new.
> > >
> > > *"I was thinking the paper tries to consider the non-i.i.d demand case and design an algorithm which performs good in this environment"*
> > >
> > > As explained in the abstract, in the introduction of the paper, as well as in our rebuttal: the goal of this paper is to provide an algorithm with *theoretical guarantees* in a specific setting (non-iid demands) where no such algorithm exists so far. Please acknowledge that this is a theoretical paper.
> > >
> > > *"DDM performs even better than the one proposed in this paper in setting 4"*
> > >
> > > Indeed, and our performance with respect to DDM is discussed in Section 4. It is indeed a non-iid setting but this environment also features lost sales, non-perishablity, capacity constraints and a large number of products. DDM has been specifically designed to handle these features while providing theoretical guarantees only in the iid setting. So, in Setting 4, MaxCOSD has proven guarantees whereas DDM has not (see the discussion with Reviewer 5kk9), but DDM performs better because of its design specific to the aforementioned features.
> > >
> > > Imagine for instance that we change Setting 4 by considering instead another feasible set $\mathcal Y$ or perishable products, then we would end up with a setting for which MaxCOSD can be applied (and still has provable guarantees) whereas, to the extent of our knowledge, no previously existing algorithm in the literature would be applicable. This fact alone should highlight the value of MaxCOSD and the significance of our results.

---

### Official Review · Reviewer_ZTux · 2023-06-23

**Soundness:** 4 excellent
**Presentation:** 4 excellent
**Contribution:** 3 good
**Rating:** 7
**Confidence:** 3

**Summary:**

The paper studies a general online decision making problem for inventory management. The problem setting extends the generic online convex optimization by introducing structures such as states and demands. Compared to settings studied by related works in inventory management, the setting of this paper relaxes several restrictive assumptions on the loss functions, dynamics and demands. Then, the problem is solved by a new batch-update online gradient descent algorithm with an optimal regret upper bound, whose analysis relies on a geometric cycle property induced by the problem structure. The strengths and limitations of this algorithm are further demonstrated through experiments.

**Strengths:**

Overall, this is a strong submission.

- It is very well-written, with a smooth and clear coverage of the motivation, how the problem connects to classical online learning models, and the novelty of the proposed algorithm.

- Related works on the inventory problem are discussed in detail. This is especially valuable since the readers are most likely from the machine learning community, who are not familiar with the necessary background.

- Although I didn't check everything in the proof, the analysis is solid to my knowledge. Intuitions of the algorithm including the nondegeneracy assumption are clearly presented and natural.

- Experiments are included, with both positive and negative results. The experimental methodology makes sense.

**Weaknesses:**

I don't have much to complain in general. One aspect I'm not sure is the novelty. The backbone of the proposed algorithm is a batch-update version of online gradient descent, which is standard from the perspective of online learning. Hence, the novelty of the paper is mainly on the combination of OGD with inventory structures, which requires sufficient domain knowledge to appreciate. Unfortunately due to my unfamiliarity with inventory problems, I cannot make a very informed evaluation. This is more of my limitation rather than the limitation of this paper. Has the geometric cycle appeared in existing analyses?

**Questions:**

- It is assumed that the adversary is oblivious, as the loss function and the demand sequence are determined at the beginning. Is there any particular reason for making this assumption? In online learning, people mainly need oblivious adversaries when the algorithm is randomized, but the algorithm in this paper is deterministic.

- The paper leaves discrete feasible sets to future works (L135). I guess this might be handled by an expert algorithm, following the similar batch-update idea. Probably out of the scope of this paper, but it seems a reasonable addition.

- The paper discusses the many ways that the setting of this paper generalizes existing works. Does the regret bound also recover existing bounds for more restrictive settings?

- Appendix D compares the algorithm to a more direct approach based on OCO. The limitation of the latter is that the competitor is subject to the feasibility constraint. This leaves me wondering, if the competitor in this paper is not feasible, then technically we cannot implement it. Then, the meaning of the competitor term $\sum l_t(y)$ becomes more obscure.

**Limitations:**

Limitations are discussed.

---

> ### Author Rebuttal · Authors · 2023-08-07
>
> 1. *One aspect I'm not sure is the novelty.* ; *The backbone of the proposed algorithm is a batch-update version of online gradient descent* ; *the novelty of the paper is mainly on the combination of OGD with inventory structures*
>
> We address this in our global response.
>
> 2. *Has the geometric cycle appeared in existing analyses?*
>
> Yes. It appears in the proof of the CUP paper: with their assumptions, the authors observe that their cycles length are geometrically distributed with a parameter lower bounded by mu, which is a particular case of what we call “geometric cycles”. We did not see this concept as very important, but it is true that it would be fair to to explain the reader that this concept appeared before (even if unnamed). We will fix that.
>
> 3. *It is assumed that the adversary is oblivious [...] Is there any particular reason for making this assumption? [...]*
>
> This is a good question, which we debated while writing the paper. We eventually decided to set our paper with an oblivious adversary, which we explain below.
> The first reason is that an oblivious adversary is the standard setting for online inventory problems. Indeed, fully adversarial environments make little sense from a modelling perspective. We would like to quote here the introduction of Lugosi et al. in [15], because we believe that it summarizes perfectly the situation: "*[Oblivious adversarial] is also referred to in the literature as the “adversarial” approach, although this term is somewhat unfortunate: in our setting, the market chooses a sequence of demands for the different time periods arbitrarily, as far as the inventory manager is concerned. Importantly though, the market does not adapt its strategy according to the actions of the inventory manager. Indeed, it seems far-fetched to assume that an entire market adapts, and acts adversarially, to the inventory decisions of a firm. So, while our modeling framework can capture demand correlations and nonstationarities to a significant extent, it is certainly not a game theoretic model.*"
> The second reason is related to the notion of regret we want to use if the adversary is not oblivious. Let us consider for instance an "adaptive" adversary, in the sense precisely defined in *Online bandit learning against an adaptive adversary: from regret to policy regret* from Arora et al.. More precisely, it is a setting where the functions are chosen in advance but depend on all the past decisions. In such adaptive setting, we could use the classical notion of regret, but it would be more difficult to interpret. Indeed, the comparator (the constant strategy) would be evaluated against the losses chosen in reaction to the previous choices made by our algorithm. An other option would be to consider a policy regret (suggested by reviewer 5kk9)(note that in our oblivious setting policy regret is equal to regret). Policy regret makes more sense from a modelling perspective, but the counterpart is that it makes the analysis more difficult, and somehow requires to make some extra assumption (see for instance Theorem 1 in Arora et al.). We do not want to make superfluous assumptions on our problem, so we would need to make an assumption limiting how adaptive the adversary can be. We found one way of doing this: consider that the losses depend not only on y_t but also on the state x_t, in a very specific way. Precisely, assume that the losses have the form f_t(y,x) = L_t(y) + delta(y,x), where delta(y,x) is equal to 0 if y \preceq x, +\infty otherwise. Forcing the losses to have this form can be seen as a way to restrict the adaptive adversary. We observe that this is exactly equivalent to our current setting: a sequence y_t is feasible and has a certain classical regret bound if and only if the policy regret with those losses have the same bound. This is due to the fact that those new losses f_t are finite and equal to L_t if an only if the feasibility constraint is satisfied. We considered that it was not worth doing it, as it would introduce more technicalities in a paper for no apparent gain.
>
> 4. *The paper leaves discrete feasible sets to future works (L135). I guess this might be handled by an expert algorithm, following the similar batch-update idea. Probably out of the scope of this paper, but it seems a reasonable addition.*
>
> See our global response in the section about discrete sets (in short: it is a good idea but not trivial to apply because of the feasibility constraints).
>
> 5. *The paper discusses the many ways that the setting of this paper generalizes existing works. Does the regret bound also recover existing bounds for more restrictive settings?*
>
> See our global response (in short: yes for the dependency on T, more or less for mu)
>
> 6. *Appendix D compares the algorithm to a more direct approach based on OCO. The limitation of the latter is that the competitor is subject to the feasibility constraint. This leaves me wondering, if the competitor in this paper is not feasible, then technically we cannot implement it. Then, the meaning of the competitor term [...] becomes more obscure.*
>
> In our setting the competitor is always feasible (see our explanations in the global response).
> This is why our approach makes sense, while the one described in Appendix D.1 is problematic.

---

> > ### Comment · Reviewer_ZTux · 2023-08-11
> >
> > Thanks for your detailed and particularly straight-to-the-point response. It makes good sense to me, and I'll continue voting for acceptance.

---

### Official Review · Reviewer_UtoR · 2023-07-07

**Soundness:** 4 excellent
**Presentation:** 3 good
**Contribution:** 3 good
**Rating:** 6
**Confidence:** 3

**Summary:**

The paper introduces the online inventory optimisation problem. In the problem, we have $n$ products to stock and every day choose the levels to stock up the products to. We then suffer loss according to a convex loss function, which may vary from day to day (e.g., it may depend on the demand). We also get to know a subgradient of the loss. After the demand has wiped out some of the stock, we need to choose new levels to top up the remainders too etc.

We want an online algorithm with a guarantee on the regret w.r.t. all possible constant levels.

The paper proposes an algorithm, which is an extension of the online subgradient descent (OSD).

OSD would have been applicable to the problem directly, if not for the presence of remainders (we cannot top up to a level below the current remainder). The version of the protocol without remainders is called stateless in the paper, so the proposed algorithm is a generalisation of the OSD to the "stateful" setup.

The paper proceeds to obtain a $O(\sqrt{T})$ regret bound for the algorithm.

**Strengths:**

I think the paper formulates an important problem and obtain a useful result.

**Weaknesses:**

The algorithm and the pound in the paper are an extension of the online subgradient descent (OSD). One can argue that the result is of the incremental nature.


**Questions:**

A constant strategy will only work if the constant is above the remainders all the time; otherwise it makes no sense. Am I correct that the regret is w.r.t. the constant strategies that work in this sense?

**Limitations:**

Yes

---

> ### Author Rebuttal · Authors · 2023-08-07
>
> 1. *The algorithm and the pound in the paper are an extension of the online subgradient descent (OSD). One can argue that the result is of the incremental nature.*
>
> Our answer : we address this in the global response.
>
> 2. *A constant strategy will only work if the constant is above the  remainders all the time; otherwise it makes no sense. Am I correct that  the regret is w.r.t. the constant strategies that work in this sense?*
>
> Our answer : we address this in the global response (in short: our regret is always well defined in our context).

---

### Official Review · Reviewer_5kk9 · 2023-07-09

**Soundness:** 3 good
**Presentation:** 3 good
**Contribution:** 2 fair
**Rating:** 7
**Confidence:** 4

**Summary:**

The paper considers a general family of online inventory management problems under convex losses and stateful inventory dynamics. They propose a general algorithm based on an adaptation of ideas from online convex optimization. The method is essentially AdaGrad enhanced with an additional delayed update mechanism that ensures that the inventory constraints are always satisfied. The main result is a sublinear regret guarantee against the best fixed base-stock policy chosen in hindsight, given full knowledge of the realization of the demand sequence.



**Strengths:**

The paper is very well-written when it comes to both its informal and technical content. In particular, the authors provide an excellent and succinct introduction into the relevant part of the management / operations literature, and discuss their results in this context extensively. The algorithm itself is natural and effective.

**Weaknesses:**

While I generally like the paper, there are a couple of things that I would like to have some more clarity on before I can fully support its acceptance:
- Regret is defined against the "best feasible constant strategy", as defined in Eq.(3). I worry that this comparator strategy is problematic in the stateful setting. First off, some fixed constant strategies may be infeasible when some of the carryover inventory levels x_t are large. Thus, the best feasible constant strategy may be a very poor one, ordering very high levels to make sure that the constraint is never violated. Second, I am concerned that evaluating a fixed policy on the loss sequence generated in response to the online agent's actions may be meaningless in the first place, and it might be more sensible to consider "policy regret" by taking into account the fact that the environment would respond differently to the comparator policy than to the agent. Treating this case, however, seems much harder than what is being handled in the present paper. Can the authors let me know if I'm missing something here?
- The assumption that the order levels are continuous seems like a very important one. The authors themselves admit that this is a limitation of their method. However, I tend to believe that it is a rather major limitation, as just about every practically relevant problem instance in this space features discrete actions. In lack of continuity, one has to deal with the challenge of partial information due to lost sales, which makes it impossible to estimate the gradients even after convexifying the action space via randomization (as noted by Huh and Rusmevichientong [9]). It would be useful to clarify this to the reader so they don't end up believing that this assumption is minor.
- I find it a bit disappointing that the regret bounds depend so heavily on the demand parameters mu and rho. I appreciate the negative results showing that certain degenerate demand sequences can lead to linear regret, but it's not clear to me if this justifies the poor linear dependence on 1/mu and 1/rho demonstrated by the bound. How do we know that this dependence cannot be improved to log(1/mu) and log(1/rho) or something even better? I really appreciate the experiments that plot performance as a function of the learning rate gamma, but I would also like to understand which values of gamma actually satisfy the constraint required by the theorem --- I suspect only the very very small ones, for which the algorithm works rather poorly. In this sense, it may be unfair to criticize DDM for not having performance guarantees since MaxCOSD doesn't have any either for the majority of the studied stepsizes. This is of course always a problem for theoretically motivated stepsizes, but it would be important to mention it at least once in passing. (A final note on the experiments: my impression was that AIM was also a very similar policy to what is being proposed here, but see no discussion on this.)

Overall, I think this could be a good addition to the program, but I cannot confidently recommend acceptance until the authors respond to the above concerns.

**Questions:**

see above

---

> ### Author Rebuttal · Authors · 2023-08-07
>
> 1. *I worry that this comparator strategy is problematic in the stateful setting. First off, some fixed constant strategies may be infeasible when some of the carryover inventory levels x_t are large. Thus, the best feasible constant strategy may be a very poor one, ordering very high levels to make sure that the constraint is never violated.*
>
> We address this in our global response (in short: constant strategies are always feasible).
>
> 2. *I am concerned that evaluating a fixed policy on the loss sequence generated in response to the online agent's actions may be meaningless in the first place, and it might be more sensible to consider "policy regret" by taking into account the fact that the environment would respond differently to the comparator policy than to the agent. Treating this case, however, seems much harder than what is being handled in the present paper. Can the authors let me know if I'm missing something here?*
>
> With our assumptions on the model (see remark 4) the adversary/environment is oblivious. Therefore, the regret and policy regret coincide, since the losses are oblivious and consist of functions of the current order-up-to level only. The loss functions do not change according to our decisions (oblivousness), only the states and thus the feasibility constraint vary. We will update remark 4 to make this clear to the reader. For more discussion on this topic, see maybe our answer number 3 to Reviewer ZTux, which asks us why we chose the environment to be oblivious.
>
> 3. *The assumption that the order levels are continuous seems like a very important one [...] It would be useful to clarify this to the reader so they don't end up believing that this assumption is minor.*
>
> See our global response (in short: we agree)
>
> 4. On the rates: *I find it a bit disappointing that the regret bounds depend so heavily on the demand parameters mu and rho [...] How do we know that this dependence cannot be improved to log(1/mu) and log(1/rho) or something even better?*
>
> See our global response (in short: we do not know, and other papers obtain similar constants)
>
> 5. On the parameter $\gamma$: *I really appreciate the experiments that plot performance as a function of the learning rate gamma, but I would also like to understand which values of gamma actually satisfy the constraint required by the theorem --- I suspect only the very very small ones, for which the algorithm works rather poorly. In this sense, it may be unfair to criticize DDM for not having performance guarantees since MaxCOSD doesn't have any either for the majority of the studied stepsizes. This is of course always a problem for theoretically motivated stepsizes, but it would be important to mention it at least once in passing.*
>
> We agree with the reviewer. It is true that we run MaxCOSD for a wide range of parameters which might violate our condition $\gamma \leq \rho/D$ in Theorem 12. For the record, in our experiments we always have $\rho = 1$ and $D \simeq 10,10,10^3,10^4,10^4$ for Settings 1-5 (in this order), while we explore the interval $[10^{-5}, 10^1]$. From this perspective, MaxCOSD lacks theoretical guarantees when using the parameters achieveing the best performance  (around $10^{-1}$) for Settings 3-4-5. This being said, we observe that we can force Theorem 12 to accept large stepsizes, by taking rho as large as needed. This would impact our Assumption 10 on the demand, and for things to work we would need to assume that there is a nonzero probability for all products to have a very high demand. This is either unrealistic in practice, or would hold with such a small mu that the rates would be meaningless.
> To correct this, we propose to ponder our sentence "*Remember that in this setting the baseline DDM has no whatsoever theoretical guarantees*" with the information that MaxCOSD also violates our theory as well for large stepsizes.
>
> 6. *A final note on the experiments: my impression was that AIM was also a very similar policy to what is being proposed here, but see no discussion on this.*
>
> We respectfully disagree. Our policy is quite similar to CUP's, which we discuss in Section 4. Instead, AIM performs projections onto the feasibility constraint, and do not involve cycles as we do. We agree that they share the same skeleton based on OSD, but from our perspective AIM is a rather different algorithm.

---

> > ### Comment · Reviewer_5kk9 · 2023-08-15
> >
> > Thank you for the detailed and honest response! I really appreciate the effort put into the rebuttal and in particular the detailed answers given to all my questions. Let me respond in detail:
> >
> > 1. Re feasibility of constant comparators: got it, thanks!
> > 2. Re policy regret: thanks for the clarification! I guess I should have been more explicit about my concern here. What I worried about is that the losses may implicitly depend on the carryover state $x_t$, and it may be meaningless to compare the total loss of the learner with that of a comparator strategy that would have generated a different sequence of states. I now see that this is explicitly disallowed by the model. I am not super familiar with the inventory management literature so I am not sure if it makes sense to consider losses that depend on $x_t$ --- in my mind, it may make sense to model the cost of storing leftovers between order periods with such dependence.
> > 3. Re continuous order levels: thanks for acknowledging the importance of this assumption!
> > 4. Re lower bounded demands: OK, fair enough. I am still unsure about the importance of lower-bounded demands. E.g., Lugosi et al. [15] do not require such assumptions and still get good rates in a related setting.
> > 5. Re parameter settings for the experiments: OK, that makes sense, please do add this clarification to the final version.
> > 6. Re the relationship with AIM: sorry for phrasing this comment like that. I did not want to suggest that you failed to compare to AIM, but rather ask for a comment on the relationship (in particular any dis-/ similarities) between the proposed method and AIM.
> >
> > Overall, I am happy to recommend acceptance and have raised my score accordingly.

---

> > > ### Author Response · Authors · 2023-08-16
> > > **Answer to the comments by Reviewer 5kk9**
> > >
> > > We thank the reviewer for their feedback and for taking the time to participate to the discussion.
> > >
> > > Here are our answers to some of the comments:
> > >
> > > 2: Re losses depending on x_t: The reviewer's intuition is correct, it makes sense to consider losses depending on the inventory state x_t. Such losses appear in the literature, and fall outside of our model. We explain this limitation at the end of Remark 7 (on the losses). To make clear what exactly we are losing, we can say that 1) for lost sales dynamics, losses in x_t are often equivalent (up to a reparametrization) to losses in y_t. This is the case of the newsvendor loss, which is the mostly considered one in practice. This is due to the simple relation between x_t and y_t. 2) for dynamics involving perishability, the relation between x_t and y_t is more complicated, and there it can be natural to have losses depending in x_t (see e.g. [32] which considers an outdating cost).
> > >
> > > 4: Re the importance of lower bounded demands. Allow us to point out that [15] obtains optimal rates without this assumption because they are in a *stateless* setting. As we explain in Sections 3.1 and 5, a stateless setting does not require any assumption on the demand (which is usual for OCO). Our consideration of a more general  dynamic forces us to make an assumption on the demand (as established in Proposition 13). That said, one could argue that our assumption can be replaced by a weaker one. But because our competitors need the same/similar assumption in their more specific settings (see below Assumption 10) we feel like replacing it will not be an easy task.
> > >
> > > 6: Re the relationship with AIM: thanks for further precising your comment. What we can do is to complete the paragraph following Algorithm 2, where we compare our method to CUP. We could add a sentence explaining that our method (and CUP) handle the feasibility constraint thanks to their cycles, which differs from the approach in AIM/DDM where projections onto the constraint are performed (we already explain in Section 3.2 that AIM is based on a subgradient descent and dynamic projection onto the feasibility constraint).

---

### Author Rebuttal · Authors · 2023-08-07

We will use this general response to address issues which were raised simultaneously by several reviewers. We responded to all the other issues in the individual responses to reviewers.

1) *Reviewers 5kk9, UtoR, ZTux raised concerns on the fact that the constant competitor in the regret definition might not staisfy the feasibility constraint.*

Our answer : There is no issue, because constant strategies are always feasible (we give a short proof below), so our regret is well-defined. This should absolutely appear in the paper, and we propose to state it clearly after Definition 3 (defining the regret), eventually pointing to a formal proof in the Appendix D.

The proof: Given any fixed order-up-to level $y\in\mathcal Y$, consider the constant strategy $y_t=y$ for all $t\in \mathbb{N}$, and denote by $x_t$ the states associated to this strategy. First of all, $y_1$ is feasible because we impose $x_1=0$. Then, for every $t\in\mathbb N$ we can write that $y_{t+1} = y \succeq [y-d_t]^+ = [y_t-d_t]^+ \succeq x_{t+1}$. The first inequality comes from the monotonicity of the positive part, while the last inequality is the inventory dynamical constraint.

2) *Reviewers 5kk9, ZTux point out that our action space is continuous instead of discrete.*

Our answer : Indeed, in our work, it is a crucial assumption that the feasible set is continuous. To handle the discrete case, two main ideas appear in the literature of online inventory problems. First, probabilistic rounding (see e.g. Huh and Rusmevichientong [9, Section 3.4] or Shi et al. [25, Section EC.4.2]) which, as mentioned by Reviewer 5kk9, even in the simple newsvendor case, requires additional knowledge, such as lost sales indicators, since a single subgradient may not provide enough information. Second, the use of expert algorithms as suggested by Reviewer ZTux, which has already been applied in stateless settings (see e.g. Lugosi et al. [15]). To adapt the cycle update technique to expert algorithms in stateful settings, we face the following challenge: how to bound efficiently the distance between consecutive (randomized) iterates? Answering this question is the key to design a sufficient condition for feasibility (as in our Lemma 15).

We propose to rewrite the end of Remark 6 (on the feasible set) to highlight the fact that not being able to handle discrete sets is a major restriction of our model. We will further complete the conclusion by developing the above mentioned ideas to tackle this issue.

3) *Reviewers 5kk9, ZTux, 8ZLu made comments about the optimality of our regret rates, and their dependency in mu and rho.*

Our answer : In Section 3 we explain that all previous works obtain a $O(\sqrt{T})$ regret, and that it is somehow optimal for this class of problems. In Theorem 12 we obtain the same sublinear regret, but it is true that we do not make the connection with this literature review, and that we never discuss the constants appearing in the bound (which is roughly D^2G/(mu rho)). To the question of whether this constant is optimal, we have absolutely no answer. We were not able to find a worst-case example where the regret has a constant lower bounded by 1/mu. We are also not aware of any result in the literature with a better dependency in mu and rho. For instance, the CUP paper has more or less the same rates as ours. If we combine [32, Theorem 2 & Remark 3 & Assumption 1], we see that their  multiplicative constant is also GD^2/(mu rho). There is also the rates for AIM, where the multiplicative constant is explicited in [9, end of the proof of Theorem 2: equation 9]. It is a bit hard/unfair to compare to ours, because they directly make the assumption that $\mathbb{E}[d]$ is bounded below, and they obtain a constant which scales like $\mathbb{E}[d]^{-6}$. If we take the liberty to say that $\mathbb{E}[d]$ scales like $\rho\mu$ (vaguely justified by the fact that $\mathbb{E}[d] \geq \rho \mu$), then we could say that their constant scales like $(\rho\mu)^{-6}$. This last discussion is imprecise and hides a lot of other constants under the carpet, but we are confident to say that AIM doesn't depend logarithmically in $1/\mathbb{E}[d]$.

To make all this clear to the reader, we propose to add after Theorem 12 a few sentences recalling 1) that our rate recovers the same $O(\sqrt{T})$ scaling than other methods presented in Section 3 and 2) that we do not know if the scaling in $\mu, \rho$ is optimal or if it can be improved, also pointing to the above mentioned references.

4) *Reviewers UtoR, ZTux, 8ZLu raise concerns about the novelty of the paper*

Our answer : This topic is partially subjective, so we will try to keep our answer factual.
We believe that we are transparent about the nature of MaxCOSD in the paper (see the introduction where we present "*MaxCOSD, which can be seen as a generalization of the Online Subgradient Descent method.*"). We also make clear that OSD naturally solves online inventory problems in the stateless setting (Section 3.1 or Corollary 20). On the other hand, we would like to emphasize that applying naively OSD to stateless inventory problems does not work (see Section D.1 in the appendix). Even under strong assumptions on the demand, one needs to understand the link between stepsize and feasibility, which is what we explain around Theorem 17 and Lemma 15.
We would also like to stress that our contribution is not limited to Theorem 12 (which can be seen technically as a fine control on the size of an adaptively chosen batchsize for OSD), but also to the contents of Section 5. Our lower bound (Proposition 13) is also new, and formalizes an idea which was in the air (but not proven).
Aside from this, we humbly hope that this paper will serve as a friendly introduction to inventory problems for the online learning community, motivating them to apply their techniques to a field where a lot of progress can still be made.

---

### Decision · Program_Chairs · 2023-09-21

**Decision:**

Accept (poster)

**Comment:**

A good paper that provides an inventory control algorithm with theoretical guarantees that also perform well empirically. The ML community will appreciate the value of its power being leveraged in the inventory control application.